# SP-MoMamba: Superpixel-driven Mixture of State Space Experts for Efficient Image Super-Resolution

## Abstract

The state space model (SSM) has garnered significant attention recently due to its exceptional long-range modeling capabilities achieved with linear-time complexity, enabling notable success in efficient super-resolution. However, applying SSMs to vision tasks typically requires scanning 2D visual data with a 1D-sequence form, which disrupts inherent semantic relationships and introduces artifacts and distortions during image restoration. To address these challenges, we propose a novel SP-MoMamba method that integrates SSMs with the semantic preservation capability of superpixels and the efficiency advantage of Mixture-of-Experts (MoE). Specifically, we pioneer the use of superpixel features as semantic units to reconstruct the SSM scanning method, proposing the Superpixel-driven State Space Model (SP-SSM) as a basic building block of SP-MoMamba. Furthermore, we introduce the Multi-Scale Superpixel Mixture of State Space Experts (MSS-MoE) scheme to strategically integrate SP-SSMs across scales, effectively harnessing the complementary semantic information from multiple experts. This multi-scale expert integration significantly reduces the number of pixels processed by each SSM while enhancing the reconstruction of fine details through specialized experts operating at different semantic scales. This framework enables our model to deliver superior performance with minimal computational overhead.

## 1 Introduction

Single-image super-resolution (SR) is a pivotal technique in image processing, aimed at reconstructing high-resolution (HR) images from their low-resolution (LR) counterparts to enhance image detail and visual quality. This technology finds widespread application across diverse fields, including medical imaging, surveillance systems, and satellite imagery. Numerous studies have leveraged convolutional neural networks (CNNs) Dong et al. (2015); Lim et al. (2017); Zhang et al. (2018) and Transformer Liang et al. (2021); Li et al. (2023b); Zhou et al. (2023) to learn this inherently ill-posed mapping relationship. However, most SR methods Lim et al. (2017); Zhang et al. (2018) relied on deeper and more complex architectures to achieve superior performance. These methods often entail high computational complexity, rendering real-time processing impractical on resource-constrained devices and thereby limiting their deployment and widespread adoption in real-world scenarios. Although some researchers have reduced computational complexity through methods such as neural architecture search Chu et al. (2021), recursive networks Tai et al. (2017), and model distillation Liu et al. (2020); Hui et al. (2018), these efforts have not yet fully addressed this issue.

Recently, state space models (SSMs), exemplified by Mamba Gu & Dao (2023), have opened new avenues for Efficient SR. Mamba offers linear computational complexity and excels at modeling long sequences, initially proving its value in high-level vision tasks such as image classification Liu et al. (2024); Zhu et al. (2024) and object detection Zhang et al. (2025a); Wang et al. (2024c). On this basis, researchers adapted Mamba for low-level vision tasks like image denoising Guo et al. (2024), image SR Qiao et al. (2024), and low-light image enhancement Zou et al. (2024); Zhen et al. (2024). For example, MambaIR Guo et al. (2024), based on visual SSM framework, achieved reconstruction quality comparable to transformer-based methods while maintaining lower computational costs. These developments demonstrate that Mamba effectively balances performance and

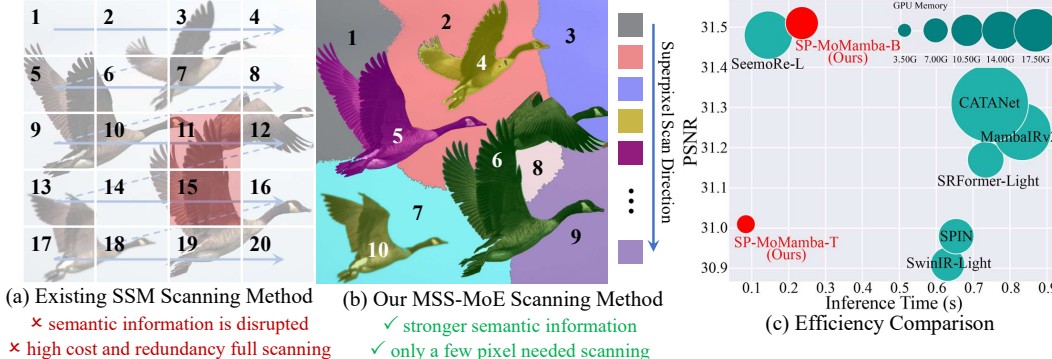

Figure 1: (a) The existing method Guo et al. (2024) suffers from the adverse effects of the scanning method of Mamba (the multi-directional scans are not shown for presentation clarity). (b) The proposed MSS-MoE scanning method can efficiently model the global information by a mixture of experts at different scale, and embed the semantic consistency of superpixels into Mamba. (c) Comparison between performance vs Inference times and GPU Memory on Manga109 ×4 dataset. Inference times and GPU Memory are calculated on 720p HR image.

efficiency in Efficient SR, though additional optimizations are still required to tailor it to specific use cases and achieve an ideal performance-efficiency trade-off.

The main challenge of Mamba-based SR methods currently is the inability to maintain correct semantic relationships during global image scanning. **Specifically, these methods convert 2D images into 1D sequences during the scanning process, which disrupts inherent semantic information and impairs SR model performance.** As shown in Figure 1 (a), this unfolding process destroys the semantic connections between spatially adjacent pixels (e.g., vertically neighboring pixels), hindering the model's ability to capture local structural details effectively. Although strategies such as multi-directional scanning Guo et al. (2024) or cascaded Mamba Qiao et al. (2024) modules attempt to mitigate this issue, they fail to address the fundamental problem of semantic disruption, instead exacerbating computational overhead and parameter complexity. Furthermore, **repetitive textures in natural images, such as skies and water surfaces, are prone to semantic confusion in 1D sequences, weakening the model's grasp of overall image structure.** This shows that there is still much room for improvement in the processing efficiency and semantic preservation of the current Mamba-based method.

To address the above challenges, we propose SP-MoMamba, an efficient SSM tailored for efficient SR. Given that superpixel features naturally delineate distinct semantic regions, as shown in Figure 1 (b), our core innovation lies in integrating their semantic preservation capabilities into a SSM framework. Technically, our SP-MoMamba is composed of stacked Layers of Experts (LoEs) for dynamically selecting the pivotal features via experts, focusing on two different aspects. At the macro level, each LoE contains two consecutive expert blocks: (a) *Superpixel Global Modulating Expert* (SGME), which excels in modeling global semantic information, and (b) *Local Spatial Modulating Expert* (LSME), which is proficient in efficient reconstruction of local spatial details. At the micro level, we design a Multi Scale Superpixel Mixture of State Space Experts (MSS-MoE) as the foundational component of SGME, which dynamically selects the optimal scale of superpixel-driven state space model (SP-SSM) for different inputs at different scales, to accurately capture the correlation between global semantics. Specifically, SP-SSM compresses semantically homogeneous features into superpixel units through superpixel sampling. Then, SSM calculates the similarity between superpixels. This similarity is propagated to the corresponding semantic regions to enhance consistency within the regions. Overall, our method obtains different professional knowledge by explicitly mining experts of different granularity for different expertise, thereby accurately reconstructing more details. Our contributions are summarized below:

- To the best of our knowledge, SP-MoMamba is the first work that pioneers the use of superpixel features as fundamental semantic units to restructure the input for State Space Models (SSMs). We correspondingly introduce a Superpixel-driven State Space Model (SP-SSM), which effectively resolves the issue of semantic disruption inherent in the scanning process of Mamba-based methods.

- We propose a Multi-Scale Superpixel Mixture of State Space Experts (MSS-MoE), enabling comprehensive global modeling by dynamically selecting optimal experts across scales to leverage semantic similarities.
- Quantitative comparisons in Figure 1 (c) further confirm the advantages of our method: it surpasses other efficient SR techniques in reconstruction fidelity and achieves a significant reduction in inference time.

## 2 RELATED WORKS

**Efficient Super-resolution Methods.** Efficient super-resolution methods have been pursued through lightweight architectures Ahn et al. (2018); Hui et al. (2018); Sun et al. (2023) and efficient transformers Zhang et al. (2022); Lu et al. (2022); Zou et al. (2022). CARN Ahn et al. (2018) uses cascading for feature integration, IMDN Hui et al. (2018) employs feature distillation, and SAFMN Sun et al. (2023) builds channel-aware pyramids. To reduce Transformer complexity, methods like ELAN Zhang et al. (2022) and ESRT Lu et al. (2022) lower dimensionality, while SCET Zou et al. (2022) uses pixel attention. Recently, SPIN Zhang et al. (2023) leverages superpixel and cross-attention. However, balancing efficiency and performance remains challenging.

Recently, Mamba Gu & Dao (2023), a selective SSM, has been successfully adapted to the vision domain, such as VMamba Liu et al. (2024) and VIM Zhu et al. (2024). Subsequently, its application has been further explored in low-level vision tasks, yielding a variety of methods Zhen et al. (2024); Zou et al. (2024) with promising results. MambaIR Guo et al. (2024), for instance, captures spatial information and enhances channel interactions. However, these Mamba-based methods rely on multi-directional scanning to process all pixels, disrupting semantic coherence and increasing computation. In contrast, our approach uses superpixels to extract compressed semantic features, modeling their spatial relationships with SSMs, preserving semantics while significantly reducing complexity.

**Mixture of Experts (MoE).** Recently, the Mixture of Experts (MoE) method has gained widespread adoption in large-scale language models due to its efficiency and scalability. Thus, MoE has been extended to advanced vision tasks, including image classification Riquelme et al. (2021), object detection Wu et al. (2022), as well as low-level vision tasks Emad et al. (2022); Zamfir et al. (2024); Rossi et al. (2025). For example, literature Emad et al. (2022) and Liang et al. (2022) extract latent degradation features to construct MoE-based adaptive networks, effectively addressing diverse degradation patterns in blind SR. SeemoRe Zamfir et al. (2024) employs rank-modulated experts to prioritize features with the highest information content, followed by spatial modulation experts to achieve precise spatial enhancement. Similarly, Swin2-MoSE Rossi et al. (2025) enhances Swin2SR Conde et al. (2022) by incorporating an MoE framework, yielding improved visual outcomes. While these methods leverage the flexibility and efficiency of MoE to achieve commendable performance, there remains potential for further improvement in image quality.

## 3 MOTIVATION

To extend state space models (SSMs) from 1D sequence data to 2D visual data, most current research Guo et al. (2024); Qiao et al. (2024); Liu et al. (2024) employs a 2D selective scanning mechanism (SS2D) Liu et al. (2024) to capture spatial correlations, as illustrated in Figure 1(a). However, flattening an image into a 1D sequence often disrupts inherent semantic relationships. For example, two geese that are spatially adjacent in Figure 1(a) might end up widely separated in the 1D sequence, hindering the model's ability to leverage their proximity for semantic inference. Furthermore, images frequently contain repetitive structures like skies and buildings which share similar textures, heightening the risk of semantic confusion. Once unfolded, information from these structures may be incorrectly associated, leading to erroneous predictions. Thus, current SS2D methods struggle to adequately preserve critical spatial structure and semantic information.

Compared to traditional SS2D, which systematically transforms 2D features into 1D sequences, superpixel sampling clusters semantic similar pixels based on color or texture. This effectively reduces the number of pixels requiring processing while maintaining the image's spatial structure and semantic integrity. Therefore, integrating superpixel algorithms into SSMs provides a robust solution to the limitations of conventional SSMs when processing 2D images.

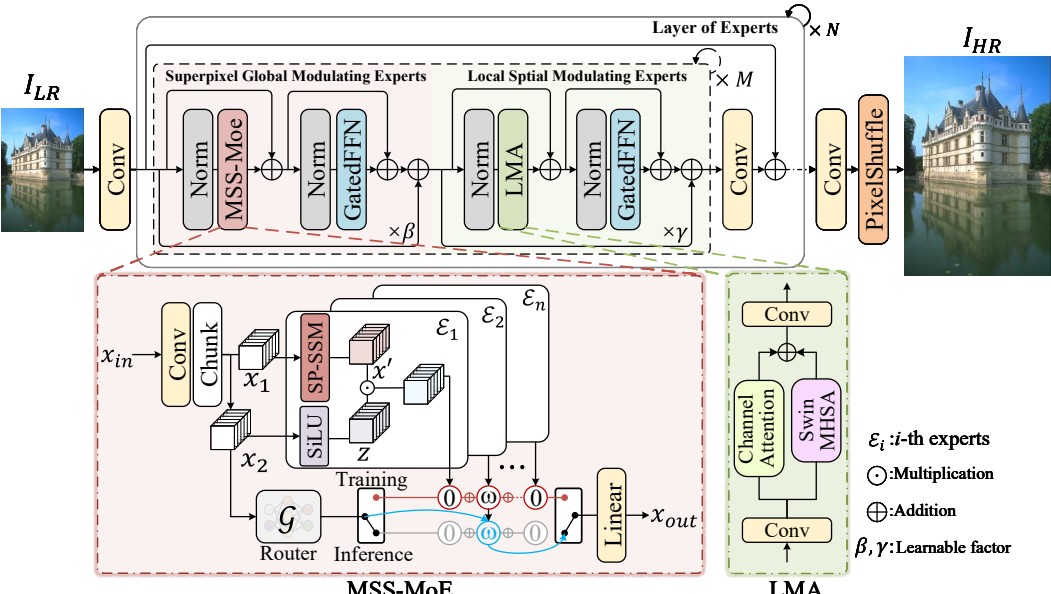

Figure 2: Architecture of our proposed method. SP-MoMamba is composed of several Layer of Experts (LoEs). Each LoE is composed of a superpixel global modulation expert (SGME) and a local spatial modulation expert (LSME). SGME uses multi-scale superpixel mixture of state space experts (MSS-MOE) to select the optimal semantic correlation to refine the global texture, while LMSE uses local mixed attention (LMA) to further restore the local texture.

To verify the issues of semantic interruption in Mamba based methods, we analyze the Local Attribute Maps (LAM) and Diffusion Index (DI) presented in Figure 3. As illustrated, traditional methods like MambaIR and MambaIRv2 exhibit restricted activation areas, reflecting the limitations of standard SS2D in capturing long-range correlations within repetitive structures. In contrast, our proposed SP-MoMamba demonstrates significantly broader activation coverage, indicating the successful utilization of non-local, perceptually similar features. This visual evidence is verified quantitatively by the highest DI score of 45.53 (surpassing 43.61 and 41.46), confirming that our superpixel-integrated approach effectively enhances the model's receptive field and maintains semantic integrity. More visual analysis can be found in the Supplementary material.

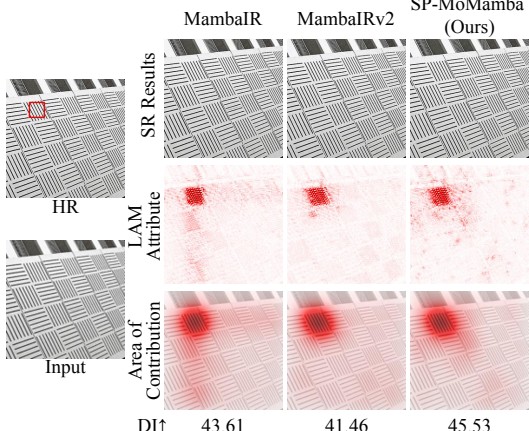

Figure 3: Analysis semantic preservation capability of different methods. The larger the Diffusion Index (DI), the more semantically similar pixels are involved in restoring the corresponding region.

## 4 METHODOLOGY

In this section, we present our proposed SP-MoMamba, as illustrated in Figure 2. The complete architecture of our pipeline integrates $N$ layers of experts (LoEs) and upsampling layers. Initially, a $3 \times 3$ convolutional operation is employed to extract shallow features from the input low-resolution image. These features are then processed through a series of LoEs to recover deep features. Each LoE consists of $M$ paired sets of Superpixel Global Modulating Experts (SGME) and Local Spatial Modulating Experts (LSME), collaboratively enhancing feature restoration. SGME adopts a collaborative reconstruction method by integrating a multi-scale superpixels mixture of state space experts (MSS-MoE), maximizing the interaction of global information. LSME concentrates on refining local features through a localized mixed attention mechanism, which enhances overall performance. In

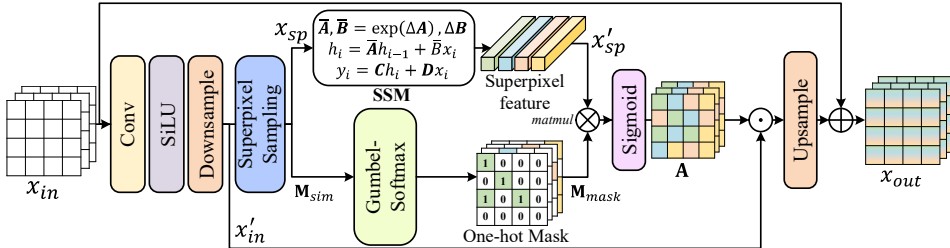

Figure 4: Illustration of the superpixel-driven state space model (SP-SSM). One-hot mask should be $N \times M$, which is converted into a 3D matrix $(H \times W) \times M$ for ease of understanding in the figure.

addition, two residual connections with learnable scales $\beta$, and $\gamma$ are introduced. Finally, the refined deep features are transformed into high-resolution images via a pixelshuffle and convolution.

## 4.1 SUPERPIXEL GLOBAL MODULATING EXPERTS

Unlike SPIN Zhang et al. (2023) and MambaIR Guo et al. (2024), which rely on substantial computational resources, we prioritize efficiency by constructing global similarity relationships based on interactions among the most relevant scale-specific superpixels. Here, we propose the MSS-MoE, as shown in Figure 2. MSS-MoE employs superpixel-driven state space models (SP-SSMs) at different scales to independently model global features across different resolutions. By leveraging the strengths of the mixture of experts scheme, it selectively integrates the resulting features, ensuring optimal global modeling within each LoE. Then, a Gated Feed-Forward Network (GatedFFN) Chen et al. (2023) is utilized to aggregate contextual information from these global features.

**MSS-MoE.** As shown in Figure 2, the output features from the last layer normalization serve as the input features $x_{in}$ for this module. A Linear layer is first applied to increase the dimensionality of the feature channels, followed by a split along the channel dimension to yield two distinct features, $x_1$ and $x_2$. Subsequently, we employ a SP-SSM module to derive the global attention feature $x'$ from $x_1$. Meanwhile, $x_2$ is processed through an activation function to obtain the gating feature $z$. Consequently, the formulation for each superpixel state space expert is expressed as follows:

$$\mathcal{E}_i(x_1, x_2, s) = x' \odot z = \text{SP-SSM}(x_1, s) \odot \sigma(x_2) \tag{1}$$

where SP-SSM$(\cdot)$ and $\sigma(\cdot)$ denote the SP-SSM module and the SiLU Shazeer (2020) activation function, respectively. The $\odot$ denotes Hadamard product. The $s$ represents the scale parameter of SP-SSM. We employ an SP-SSM to ensure robust modeling of global information while introducing an residual connection to prevent the loss of local information.

However, the SP-SSM operating at a fixed scale may fail to fully exploit all internal information, thereby limiting the model's expressive capacity. To address this, we propose an ensemble approach that integrates superpixel state space experts across multiple scales $s_i$. A routing network searches the solution space to identify the optimal scale for the superpixel state space experts based on the input and network depth. The final output $x_{out}$ of the MSS-MoE is formulated as follows:

$$x_{out} = \sum_i^n \mathcal{G}(x_2)\mathcal{E}_i(x_1, x_2, s_i) \tag{2}$$

where $\mathcal{G}(\cdot)$ and $\mathcal{E}(\cdot)$ denote the router function and the $i$-th expert function, respectively. The $s_i$ represents the scale parameter of the $i$-th expert's SP-SSM module. Specifically, a router $\mathcal{G}(\cdot)$ is composed of a linear mapping and Softmax to map input features into weights of different superpixel state space experts. The sparsity inherent in the router function $\mathcal{G}(\cdot)$ optimizes computation by assigning greater weights to the top-$k$ superpixel state space experts. During training, our method learns from all superpixel state space experts, while during inference, it utilizes only the selected top-$k$ experts with higher routing weights for computation, thereby enhancing efficiency. Hence, the computational complexity of the inference process becomes independent of the total number of experts, further enhancing efficiency. We further provide the pseudocode for the proposed MSS-MoE in the supplementary materials.

**SP-SSM.** As shown in Figure 4, given the input feature $x_{in} \in \mathbb{R}^{H \times W \times C}$, we use $3 \times 3$ convolution and SiLU activation function to map features. Then, these features are downsampled by a factor of

$s$, and superpixel sampling is performed to obtain the corresponding $M$ superpixel features $x_{sp} \in \mathbb{R}^{M \times C}$ and similarity matrix $\mathbf{M}_{sim} \in \mathbb{R}^{N \times M}$ (where $N = H \times W$). It is formulated as follows:

$$x_{sp}, \mathbf{M}_{sim} = \text{SPS}(\sigma(\text{Conv}(x_{in})) \downarrow_s) \tag{3}$$

where $\text{SPS}(\cdot)$ denotes the superpixel sampling operation. $\downarrow_s$ represents the downsampling operation by a factor of $s$. Subsequently, a SSM is employed to perform global information modeling on the superpixel feature $x_{sp}$, yielding an enhanced superpixel feature $x'_{sp} \in \mathbb{R}^{M \times C}$. The similarity matrix $\mathbf{M}_{sim}$ is transformed into a differentiable one-hot mask $\mathbf{M}_{mask} \in \mathbb{R}^{N \times M}$ using the Gumbel-Softmax Jang et al. (2016) technique applied to log probabilities, enabling the indexing of the most similar superpixel for each pixel. Then, matrix multiplication followed by a sigmoid function is utilized to derive the global attention feature $\mathbf{A} \in \mathbb{R}^{N \times C}$, as follows:

$$x'_{sp} = \text{SSM}(x_{sp}) \tag{4}$$

$$\mathbf{M}_{mask} = \text{Gumbel-Softmax}(\mathbf{M}_{sim}) \tag{5}$$

$$\mathbf{A} = \text{Sigmoid}(\mathbf{M}_{mask} \otimes x'_{sp}) \tag{6}$$

where $\text{Sigmoid}(\cdot)$ and $\otimes$ denote sigmoid function and matrix multiplication. The final output of this module is obtained by multiplying the attention feature $\mathbf{A}$ with $x'_{in}$, followed by the addition of the original transformed feature $x_{in}$, as follows:

$$x_{out} = (\mathbf{A} \odot x'_{in}) \uparrow_s + x_{in} \tag{7}$$

Since superpixels encapsulate comprehensive semantic information, the resulting output features effectively capture correlations among distinct semantics.

**Superpixel Sampling.** We follow the soft $k$-means-based superpixel algorithm in SSN Jampani et al. (2018) to perform superpixel sampling on images. Given input as $\mathbf{x} \in \mathbb{R}^{N \times C}$ (where $N = H \times W$), $M$ superpixels $\mathbf{s} \in \mathbb{R}^{M \times C}$ and similarity matrix $\mathbf{M}_{sim} \in \mathbb{R}^{N \times M}$ are obtained through $T$ iterations, maximizing their association with the corresponding pixels. Firstly, as shown in Figure 5, we use average pooling to initialize superpixels $\mathbf{s}^0$. Then, we conduct iterations using a similarity matrix that calculates the similarity between each pixel and superpixel. It can be formulated as follows:

$$\mathbf{M}^t_{sim}(i, j) = e^{-||\mathbf{x}(i) - \mathbf{s}^{t-1}(j)||^2_2} \tag{8}$$

Notably, superpixel sampling solely evaluates the similarity mapping between each pixel and

Figure 5: Superpixel sampling of our method, which initializes the superpixel features by average pooling, and then generates the superpixel features and similarity matrix.

its neighboring superpixels. This preserves the local coherence of superpixels, thereby enhancing computational efficiency. Subsequently, we can obtain the superpixel $\mathbf{s}^t$ by computing a weighted sum of all pixels, defined as:

$$\mathbf{s}^t_j = \frac{1}{\mathbf{z}^t(j)} \sum_i \mathbf{M}^t_{sim}(i, j) \mathbf{x}(i) \tag{9}$$

where $\mathbf{z}^t(j) = \sum_i \mathbf{M}^t_{sim}(i, j)$ denotes the normalization term along the column. After $T$ iterations, we can obtain the final similarity matrix $\mathbf{M}^T_{sim}$ and superpixels $\mathbf{s}^T$. Using the similarity matrix, we can assign each pixel to its most similar superpixel, thus generating the corresponding mask, as depicted in Figure 5. Therefore, with the superpixels and their respective masks, we can perform superpixel-based attention weighting on pixels across distinct regions. Our proposed SP-SSM utilizes this critical insight by employing a SSM to assign weights to superpixels, enabling weighted processing of semantic information across distinct regions.

## 4.2 LOCAL SPATIAL MODULATING EXPERTS

After the proposed MSS-MoE primarily leverages superpixels to capture global semantic relationships, we enhance its capability by incorporating Local Spatial Modulation Experts (LSME) to

strengthen the processing of local information. Given that MSS-MoE requires scanning only a limited number of superpixels, this property is insufficient for the modeling of local correlation. Consequently, we adopt a robust combination of shift window-based multi-head self-attention (SWin-MHSA) and channel attention to construct a Local Mixed Attention Module (LMA), as depicted in Figure 2. Channel attention recalibrates features across channels to emphasize salient local information; subsequently, Swin-MHSA captures fine-grained spatial dependencies within local windows. GatedFNN then refines features by integrating global and local information, preserving prior semantics while enhancing detail capture, thus improving overall model performance.

| Scale | Model | Params (M)↓ | GMACs↓ | Set5 | | Set14 | | BSD100 | | Urban100 | | Manga109 | |
|---|---|---|---|---|---|---|---|---|---|---|---|---|---|
| | | | | PSNR↑ | SSIM↑ | PSNR↑ | SSIM↑ | PSNR↑ | SSIM↑ | PSNR↑ | SSIM↑ | PSNR↑ | SSIM↑ |
| ×2 | CARN-M | 412K | 91 | 37.53 | 0.9583 | 33.26 | 0.9141 | 31.92 | 0.8960 | 31.23 | 0.9193 | — | — |
| | PAN | 261K | 71 | 38.00 | 0.9605 | 33.59 | 0.9181 | 32.18 | 0.8997 | 32.01 | 0.9273 | 38.70 | 0.9773 |
| | DRSAN | 370K | 86 | 37.99 | 0.9606 | 33.57 | 0.9177 | 32.16 | 0.8999 | 32.10 | 0.9279 | — | — |
| | SAFMN | 228K | 52 | 38.00 | 0.9605 | 33.54 | 0.9177 | 32.16 | 0.8995 | 31.84 | 0.9266 | 38.71 | 0.9771 |
| | FECAN-tiny | 107K | 24 | 38.00 | 0.9605 | 33.52 | 0.9172 | 32.14 | 0.8993 | 31.86 | 0.9252 | 38.76 | 0.9772 |
| | SeemoRe-T | 220K | 45 | 38.06 | 0.9608 | 33.65 | 0.9186 | 32.23 | 0.9004 | 32.22 | 0.9286 | 39.01 | 0.9777 |
| | SRConvNet | 387K | 74 | 38.00 | 0.9605 | 33.58 | 0.9186 | 32.16 | 0.8995 | 32.05 | 0.9272 | 38.87 | 0.9774 |
| | **SP-MoMamba-T (ours)** | 259K | 85 | **38.16** | **0.9612** | **33.81** | **0.9199** | **32.29** | **0.9011** | **32.48** | **0.9312** | **39.76** | **0.9820** |
| ×4 | CARN-M | 415K | 33 | 31.92 | 0.8903 | 28.42 | 0.7762 | 27.44 | 0.7304 | 25.62 | 0.7694 | — | — |
| | PAN | 272K | 28 | 32.13 | 0.8948 | 28.61 | 0.7822 | 27.59 | 0.7363 | 26.11 | 0.7854 | 30.51 | 0.9095 |
| | DRSAN | 410K | 31 | 32.15 | 0.8935 | 28.54 | 0.7813 | 27.54 | 0.7364 | 26.06 | 0.7858 | — | — |
| | SAFMN | 240K | 14 | 32.18 | 0.8948 | 28.60 | 0.7813 | 27.58 | 0.7359 | 25.97 | 0.7809 | 30.43 | 0.9063 |
| | FECAN-tiny | 121K | 7 | 32.08 | 0.8935 | 28.58 | 0.7809 | 27.57 | 0.7354 | 25.96 | 0.7801 | 30.33 | 0.9049 |
| | SeemoRe-T | 232K | 12 | 32.31 | 0.8965 | 28.72 | 0.7840 | 27.65 | 0.7384 | 26.23 | 0.7883 | 30.82 | 0.9107 |
| | SRConvNet | 382K | 22 | 32.18 | 0.8951 | 28.61 | 0.7359 | 27.57 | 0.7359 | 26.06 | 0.7845 | 30.35 | 0.9075 |
| | **SP-MoMamba-T (ours)** | 271K | 22 | **32.35** | **0.8970** | **28.77** | **0.7850** | **27.69** | **0.7398** | **26.40** | **0.7939** | **31.01** | **0.9160** |

Table 1: Comparison to efficient SR models. PSNR (dB ↑) and SSIM (↑) metrics are reported on the Y-channel. **Best** and second best performances are highlighted. GMACs (G) are computed by upscaling to a 1280 × 720 HR image.

| Scale | Model | Params (M)↓ | GMACs↓ | Set5 | | Set14 | | BSD100 | | Urban100 | | Manga109 | |
|---|---|---|---|---|---|---|---|---|---|---|---|---|---|
| | | | | PSNR↑ | SSIM↑ | PSNR↑ | SSIM↑ | PSNR↑ | SSIM↑ | PSNR↑ | SSIM↑ | PSNR↑ | SSIM↑ |
| ×2 | SwinIR-Light | 910K | 244 | 38.14 | 0.9611 | 33.86 | 0.9206 | 32.31 | 0.9012 | 32.76 | 0.9340 | 39.12 | 0.9783 |
| | SRFormer-Light | 853K | 236 | 38.23 | 0.9613 | 33.94 | 0.9209 | 32.36 | 0.9019 | 32.91 | 0.9353 | 39.28 | 0.9785 |
| | SPIN | 497K | 320 | 38.20 | 0.9615 | 33.90 | 0.9215 | 32.31 | 0.9015 | 32.79 | 0.9340 | 39.18 | 0.9784 |
| | CAMixerSR | 746K | 205 | 38.23 | 0.9613 | 34.00 | 0.9214 | 32.34 | 0.9016 | 32.95 | 0.9340 | 39.32 | 0.9781 |
| | FECAN-light* | 732K | 162 | 38.22 | 0.9614 | 34.01 | 0.9216 | 32.35 | 0.9017 | 32.89 | 0.9787 | 39.47 | 0.9784 |
| | Freqformer* | 870K | 229 | 38.26 | 0.9615 | 34.02 | 0.9217 | 32.34 | 0.9018 | 32.94 | 0.9353 | 39.47 | 0.9789 |
| | MambaIR-light | 905K | 334 | 38.13 | 0.9610 | 33.95 | 0.9208 | 32.31 | 0.9013 | 32.85 | 0.9349 | 39.20 | 0.9782 |
| | SeemoRe-L | 931K | 197 | **38.27** | **0.9616** | 34.01 | 0.9210 | 32.35 | 0.9018 | 32.87 | 0.9344 | 39.49 | 0.9790 |
| | CRAFT | 738K | 197 | 38.23 | 0.9615 | 33.92 | 0.9211 | 32.33 | 0.9016 | 32.86 | 0.9343 | 39.39 | 0.9786 |
| | MambaIRv2-light | 774K | 286 | 38.26 | 0.9615 | 34.09 | 0.9221 | 32.36 | 0.9019 | 33.26 | 0.9378 | 39.35 | 0.9785 |
| | **SP-MoMamba-B (ours)** | 543K | 170 | **38.27** | **0.9616** | 34.04 | 0.9219 | **32.38** | **0.9022** | 32.99 | 0.9357 | **40.18** | **0.9827** |
| ×4 | SwinIR-Light | 897K | 64 | 32.44 | 0.8976 | 28.77 | 0.7858 | 27.69 | 0.7406 | 26.47 | 0.7980 | 30.91 | 0.9151 |
| | SRFormer-Light | 873K | 63 | 32.51 | 0.8988 | 28.82 | 0.7872 | 27.73 | 0.7422 | 26.67 | 0.8032 | 31.17 | 0.9165 |
| | SPIN | 555K | 90 | 32.48 | 0.8983 | 28.80 | 0.7862 | 27.70 | 0.7415 | 26.55 | 0.7998 | 30.98 | 0.9156 |
| | CAMixerSR | 765K | 54 | 32.51 | 0.8988 | 28.82 | 0.7870 | 27.72 | 0.7416 | 26.63 | 0.8012 | 31.18 | 0.9166 |
| | FECAN-light* | 749K | 42 | 32.53 | 0.8990 | 28.89 | 0.7879 | 27.75 | 0.7421 | 26.78 | 0.8049 | 31.47 | 0.9181 |
| | FreqFormer* | 889K | 55 | **32.54** | 0.8991 | 28.89 | 0.7879 | 27.76 | 0.7425 | 26.73 | 0.8023 | 31.36 | 0.9178 |
| | MambaIR-light | 930K | 64 | 32.42 | 0.8977 | 28.74 | 0.7847 | 27.68 | 0.7400 | 26.52 | 0.7983 | 30.94 | 0.9135 |
| | SeemoRe-L | 969K | 50 | 32.51 | 0.8990 | 28.92 | 0.7881 | 27.77 | 0.7428 | 26.79 | 0.8046 | 31.48 | 0.9181 |
| | CRAFT | 753K | 52 | 32.52 | 0.8989 | 28.85 | 0.7872 | 27.72 | 0.7418 | 26.56 | 0.7995 | 31.18 | 0.9168 |
| | MambaIRv2-light | 790K | 76 | 32.51 | 0.8992 | 28.84 | 0.7878 | 27.75 | 0.7425 | 26.82 | 0.8079 | 31.24 | 0.9182 |
| | **SP-MoMamba-B (ours)** | 559K | 46 | **32.56** | **0.8992** | **28.93** | **0.7885** | **27.78** | 0.7426 | 26.76 | 0.8030 | **31.51** | **0.9210** |

Table 2: Comparison to lightweight SR models. PSNR (dB ↑) and SSIM (↑) metrics are reported on the Y-channel. **Best** and second best performances are highlighted. GMACs (G) are computed by upscaling to a 1280 × 720 HR image. * denote retraining based on equivalent experimental configuration.

## 5 EXPERIMENTS

### 5.1 EXPERIMENTAL SETTINGS

**Datasets and Evaluation.** Following the previous SR methods Liang et al. (2021); Zamfir et al. (2024), we utilize two widely-used datasets, DIV2K Timofte et al. (2017) and Flickr2K Lim et al. (2017) for training. We assess our method performance on five classical benchmark datasets for SR, Set5 Bevilacqua et al. (2012), Set14 Zeyde et al. (2010), BSD100 Martin et al. (2001), Urban100 Huang et al. (2015), and Manga109 Matsui et al. (2015). We also quantify the effectiveness of our method using the PSNR and SSIM metrics on the Y-channel from the YCbCr color space.

**Implementation Details.** To thoroughly train the proposed model, we augment the training data by randomly cropping it into 64 × 64 patches and further augment it through random rotations, horizontal and vertical flips. Consistent with Sun et al. (2022), we use the Adam Kingma & Ba

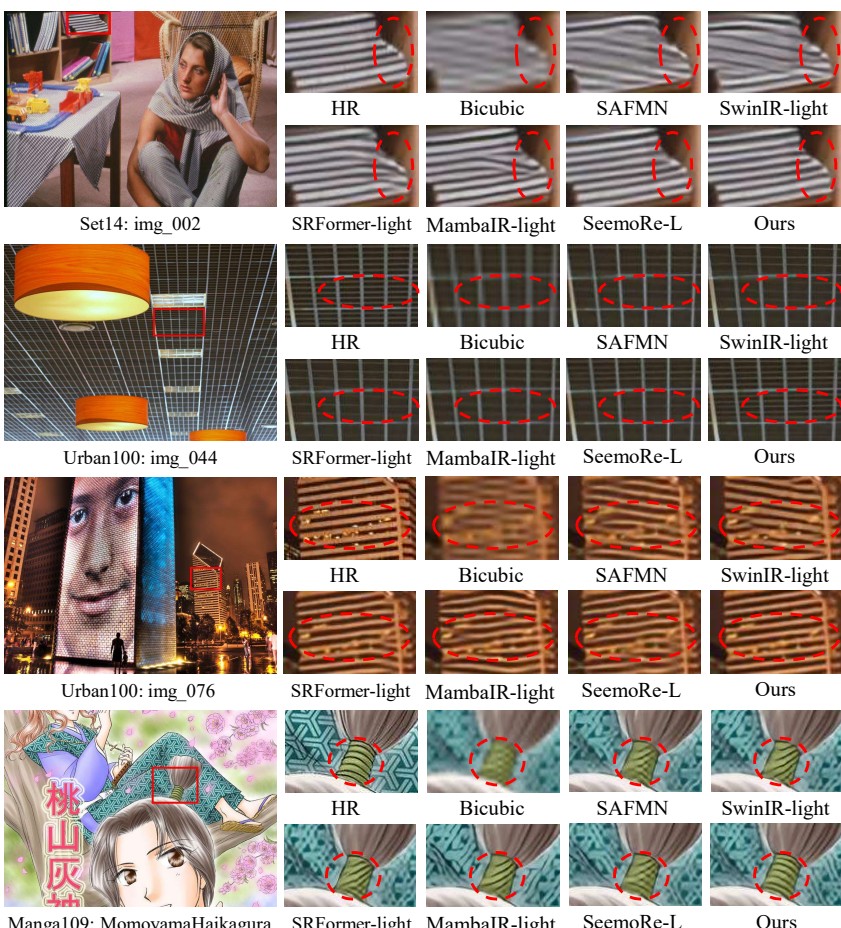

Figure 6: Qualitative comparison of our SP-MoMamba-B with state-of-the-art methods on $4\times$ SR. (Zoom in for the best view)

(2014) optimizer to minimize the $L_1$ norm between the SR output and the HR ground truth in both pixel and frequency domains across 500K iterations. The batch size is set to 32, with an initial learning rate of $1 \times 10^{-3}$ which is halved at iterations [250K, 400K, 450K, 475K]. All experiments are implemented using the PyTorch framework and conducted on a single RTX 4090 GPU. We design two variants of the SP-MoMamba model with distinct parameter configurations, denoted as SP-MoMamba-T and SP-MoMamba-B. For all MSS-MoE modules, we configure three experts with downsampling factors of [1, 2, 4]. Further details are provided in the supplementary materials.

## 5.2 COMPARISONS WITH STATE-OF-THE-ART METHODS

**Quantitative comparison.** We report quantitative results for image SR at $\times 2$ and $\times 4$ scale factors, with comparisons against current efficient state-of-the-art models presented in Table 1, including CARN-M Ahn et al. (2018), PAN Zhao et al. (2020), DRSAN Park et al. (2021), SAFMN Sun et al. (2023), FECAN-tiny Huang et al. (2025), SeemoRe-T Zamfir et al. (2024), SRConvNet Li et al. (2025b). Additionally, we evaluate against lightweight SR models such as SwinIR Liang et al. (2021), SRFormer Zhou et al. (2023), SPIN Zhang et al. (2023), CAMixerSR Wang et al. (2024b), FECAN-light Huang et al. (2025), Freqformer Dai et al. (2024), MambaIR Guo et al. (2024), CRAFT Li et al. (2025a), and MambaIRv2 Guo et al. (2025) in Table 2. Our proposed SP-MoMamba-T stands out as the most efficient method, consistently surpassing all other methods across all benchmarks and scale factors. For instance as clear from Table 1, on Urban100 and Manga109 ($\times 2$), SP-MoMamba-T exceeds SeemoRe-T Zamfir et al. (2024) by 0.26 dB and 0.75 dB, respectively. Scaling our method up to a comparable size with lightweight models yields comparable or superior results. As demonstrated in Table 2, our SP-MoMamba-B exhibits the best PSNR

| Method | SSM | SPS | MoE | LMA | GatedFFN | Params | GMACs | BSD100 | Urban100 |
|---|---|---|---|---|---|---|---|---|---|
| Baseline | - | - | - | - | - | 189K | 8.4 | 27.42 | 25.55 |
| | ✔ | - | - | - | - | 230K | 18.7 | 27.48 | 25.64 |
| | ✔ | ✔ | - | - | - | 197K | 16.7 | 27.54 | 25.86 |
| SP-MoMamba-T | ✔ | ✔ | ✔ | - | - | 242K | 16.9 | 27.59 | 26.07 |
| | - | - | - | ✔ | ✔ | 211K | 16.6 | 27.61 | 26.14 |
| | ✔ | ✔ | ✔ | - | ✔ | 261K | 19.8 | 27.63 | 26.14 |
| | ✔ | ✔ | ✔ | ✔ | ✔ | 272K | 22 | 27.69 | 26.40 |

Table 3: Ablation on key components of SP-MoMamba. We show PSNR results for ×4 upscaling.

performance on average across 5 benchmark datasets. Among them, on Manga109 (×2), our SP-MoMamba-B outperforms SeemoRe-L Zamfir et al. (2024) and MambaIRv2-light Guo et al. (2025) by 0.69 dB and 0.83 dB, respectively. As demonstrated in Figure 1(c), our SP-MoMamba strikes an optimal balance between performance and efficiency, delivering higher-quality super-resolution results than leading methods while requiring less computational time.

**Qualitative comparison.** In Figure 6, we compare the visual quality of our method against existing state-of-the-art approaches. As evident from the figure, previous methods often struggle with challenging structural textures, resulting in distortions, or inaccurate texture reconstruction. In contrast, our SP-MoMamba effectively preserves structural information and enhances clarity. For instance, in images img_044 and img_076 from the Urban100 dataset, SeemoRe-L Zamfir et al. (2024) and MambaIR-light Guo et al. (2024) fail to reconstruct the correct textures accurately. In contrast, our method can recover regular textures and complex details. These visual comparisons emphasize SP-MoMamba's effectiveness in reconstructing high-quality images by leveraging global information derived from superpixels. More visual results can be found in the Supplementary material.

## 5.3 ABLATION STUDY

We devise a set of ablation studies to evaluate the contribution and efficacy of each proposed module. All experiments are conducted on the ×4 SP-MoMamba-T setting. More ablation studies are in supplementary materials.

**Macro Architecture.** As shown in the Table 3, we assess the effectiveness of our proposed key architectural components by comparing them against a baseline model which is composed solely of residual block. The incorporation of the proposed modules into the baseline framework yields significant enhancements in performance. Specifically, the introduction of the State Space Model (SSM) and Superpixel Sampling (SPS) lays a solid foundation, where SPS effectively reduces the computational cost (GMACs) while maintaining performance. Furthermore, the addition of the Mixture-of-Expert (MoE) strategy and Local Mixed Attention (LMA) independently augments the model's capability to capture complex features. Notably, the inclusion of the Gated Feed-Forward Network (GatedFFN) alongside these components leads to the optimal configuration (SP-MoMamba-T), which augments the baseline by 0.27 dB on BSD100 and 0.85 dB on Urban100, respectively. Although the parameter count increases moderately with the integration of all modules, the substantial performance gains validate the effectiveness of each component.

| SP-MoMamba-T | #$\mathcal{E}$ | Scale | GMACs | BSD100 | Urban100 |
|---|---|---|---|---|---|
| | 1 | 1 | 22G | 27.65 | 26.17 |
| Adding experts | 2 | [1,1] | 26G | 27.67 | 26.31 |
| | 3 | [1,1,1] | 30G | 27.67 | 26.32 |
| | 4 | [1,1,1,1] | 33G | 27.68 | 26.34 |
| | 1 | 2 | 19G | 27.63 | 26.11 |
| Adding scale | 2 | [1,2] | 22G | 27.68 | 26.33 |
| | 3 | [1,2,4] | 23G | **27.69** | **26.40** |
| | 4 | [1,2,4,8] | 23G | 27.67 | 26.24 |

(a) Ablation on the number of experts and corresponding scale factor.

| Method | Params | GMACs | BSD100×4 | |
|---|---|---|---|---|
| SP-MoMamba-T | | | PSNR | SSIM |
| k=1 | 271K | 22.87 | 27.69 | 0.7398 |
| k=2 | 271K | 23.61 | 27.68 | 0.7394 |
| k=3 | 271K | 24.36 | 27.69 | 0.7398 |

(b) Ablation experiments with $k$ values on MSS-MoE. $k$ denotes the number of selected experts.

Table 4: Ablation on key components of MSS-MoE. We show PSNR results for ×4 upscaling.

**Design choices of MSS-MoE.** We explore the design choices of the MSS-MoE module by varying the number of experts and their corresponding scale factor parameters, as illustrated in Table 4(a), where #$\mathcal{E}$ denotes the number of experts. To determine the optimal number of experts and scale factors, we devise two simple scenarios: one increasing the number of experts at the same scale, and the other incorporating experts at varying scales. As evidenced in Table 4(a), employing experts across different scales outstanding enhances performance with only a modest increase in

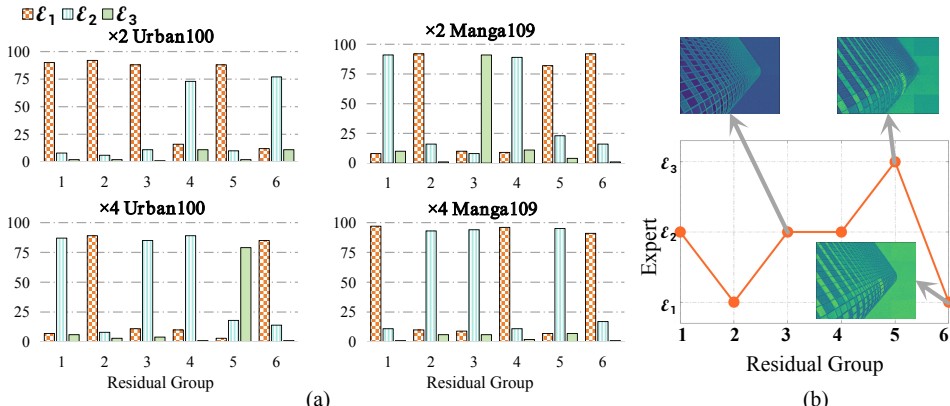

Figure 7: Expert routing analysis. (a) We plot the decisions made by the routing function for SP-MoMamba-T over the depth of the network. (b) We visualize the expert output features of SP-MoMamba-T at different scales and depth for ×4 SR given example images from Urban100.

computational cost, outperforming the method of adding experts at a uniform scale. Based on these experiments, we assert that downsampling beyond a factor of 4 results in substantial loss of semantic information, introducing much artificial information into the network and consequently degrading performance. Therefore, we select experts with scale factors of [1, 2, 4] as the final configuration. Table 4(b) analyzes the impact of parameter $k$. Results indicate that varying $k$ from 1 to 3 yields negligible differences in PSNR and SSIM, yet consistently increases computational cost (GMACs). Therefore, we select $k = 1$ as the optimal setting, achieving comparable performance with minimal complexity. These results indicate that the routing mechanism effectively leverages complementary semantic information across different scales. By dynamically selecting the optimal experts via the Router, the model significantly enhances its capability for detailed reconstruction.

## 5.4 DISCUSSION ON EXPERTS

The decision-making process of the router across different residual groups is illustrated in Figure 7(a). Notably, the network showcases a diverse range of expert choices ($\mathcal{E}_1, \mathcal{E}_2, \mathcal{E}_3$) across varying depths. This phenomenon can be attributed to the hierarchical feature learning nature of the architecture, aligning with our expectations. In fact, specific residual groups utilize larger scales ($\mathcal{E}_3$) to capture global dependencies, while others employ smaller scales ($\mathcal{E}_1$) to refine local details. Hence, static scales are less favored, as they may introduce redundancy or lack the capability to handle the varying complexity of features. This design aspect provides our method with the flexibility to adaptively adjust the processing scale, a capability that ensures efficiency. In Figure 7(b), we further visualize the routing decisions and corresponding feature maps for an exemplary input. It is noteworthy that the router intelligently switches between experts to leverage their distinct yet mutually complementary information. As the model depth increases, the network becomes proficient in restructuring these multi-scale representations for optimal restoration.

## 6 CONCLUSION

In this study, we propose SP-MoMamba, a superpixel-driven mixture of state space experts model. Unlike the other state space models, SP-MoMamba addresses the computational inefficiencies inherent in existing mamba-based restoration methods dependent on global scanning by integrating superpixel sampling with state space frameworks. This approach effectively balances robust global semantic modeling with precise local detail enhancement while minimizing complexity. In our method, the MSS-MoE module achieves comprehensive global modeling by dynamically selecting optimal experts across multiple scales, while the LSME refines local features through a synergistic combination of self-attention and channel attention mechanisms. Experimental results confirm that SP-MoMamba surpasses state-of-the-art lightweight models across various benchmark datasets.

## ETHICS STATEMENT

We confirm that we have read and committed to complying with ICLR's ethical guidelines. This study focuses on image super-resolution technology. Although it does not directly involve human subjects, we recognize that it may be misused to generate misleading content (such as deepfakes). We are committed to responsible research and hope that this technology can be applied to beneficial fields such as medical image enhancement. All experiments are based on publicly available datasets, and we have made every effort to ensure the reproducibility and impartiality of the research.

## REPRODUCIBILITY STATEMENT

To support the reproducibility of our work, we have made the following efforts. The implementation details of our proposed SP-MoMamba-T and SP-MoMamba-B architecture, including training configurations and hyperparameters, are thoroughly described in Section 5.1 and the Appendix C. The source code for the model and training scripts has been provided in our anonymized supplementary materials. Furthermore, all experiments are conducted on publicly available benchmark datasets, and the specific data preprocessing steps are clearly outlined in Section 5.1. We hope these resources will facilitate the replication of our results.

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

## APPENDIX

In this Appendix, we declare The Use of Large Language Models (LLMs). In addition, we present the principle of state space model and provide implementation details, additional experimental results and analysis.

## A    THE USE OF LARGE LANGUAGE MODELS (LLMS)

A large language model was used during the preparation of this work solely for text polishing and grammar correction. The authors are solely responsible for the entire scientific content, including all ideas, findings, and interpretations presented herein.

## B    STATE SPACE MODELS

State space models (SSMs) Gu & Dao (2023) are mathematical models used in control theory and signal processing to describe the dynamic systems. The following equation defines the standard SSMs:

$$h'(t) = \mathbf{A}h(t) + \mathbf{B}x(t) \qquad (10)$$
$$y(t) = \mathbf{C}h(t) + \mathbf{D}x(t) \qquad (11)$$

where $\mathbf{A}, \mathbf{B}, \mathbf{C}$, and $\mathbf{D}$ are the system parameters. $x(t), h(t)$ and $y(t)$ denote input, hidden state, and output. In deep learning, the existing SSM-based methods, like Mamba, employ zero-order hold (ZOH) to discretize continuous state space equations, enabling them to efficiently process long sequence data and apply to various sequence modeling tasks. It is represented by the following equations:

| Parameters | SP-MoMamba-T | SP-MoMamba-S |
|---|---|---|
| Num. RGs | 3 | 4 |
| Num. SGMEs and LSMEs | 2 | 2 |
| Channel dimension | 36 | 48 |
| MLP-Ratio | 2 | |
| Num. superpixels | 64 | |
| Num. Experts $\mathcal{E}$ | 3 | |
| Top-$k$ experts | 1 | |
| Scale list | [1,2,4] | |
| Training Dataset | DIV2K+Flickr2K | |
| Optimizer | Adam | |
| Batch size | 32 | |
| Total Num. Iterations | 500K | |
| FFT Loss weights | 0.1 | |
| LR-Rate | 1.00E-03 | |
| LR-Decay Rate | 0.5 | |
| LR-Decay Milestones | [250K,400K, 450K, 475K] | |

Table 5: Implementation Details.

$$h_t = \bar{\mathbf{A}}h_{t-1} + \bar{\mathbf{B}}x_t \qquad (12)$$
$$y_t = \mathbf{C}h_t + \mathbf{D}x_t \qquad (13)$$
$$\bar{\mathbf{A}} = \exp(\Delta\mathbf{A}) \qquad (14)$$
$$\bar{\mathbf{B}} = (\Delta\mathbf{A})^{-1}(\exp(\Delta\mathbf{A}) - \mathbf{I})(\Delta\mathbf{B}) \qquad (15)$$

where $\bar{\mathbf{A}}, \bar{\mathbf{B}}$ are discrete counterparts of $\mathbf{A}$ and $\mathbf{B}$. $\Delta$ denotes timescale parameter, which is used to convert the continuous parameters $\mathbf{A}$ and $\mathbf{B}$ into discrete ones $\bar{\mathbf{A}}$, and $\bar{\mathbf{B}}$.

## C    FURTHER IMPLEMENTATION DETAILS

Table 5 details the architectural configurations and training parameters utilized to obtain the results presented in this study. To ensure reproducibility, we maintained a consistent random seed across all experiments. Our implementation leverages the publicly available PyTorch-based BasicSR framework for both architecture design and training. Additionally, we employ the fvcore Python package to calculate GMACs and parameter counts. The pseudocode for the proposed MSS-MoE is provided in Algorithm 1.

---

**Algorithm 1** Multi-Scale Superpixel Mixture of State space Experts (MSS-MoE)

---

1: **Input:** Input feature $x_1$ and $x_2$
2: **Parameters:** $n$ experts $\mathcal{E}$, Router $\mathcal{G}$, Scale factor $s = s_1, s_2, ..., s_n$, Superpixel state space module (SP-SSM), Top-$k$ expert
3: Compute router outputs: $g = \mathcal{G}(x_2)$
4: Normalize weights: $w = \text{Softmax}(g)$
5: Select top-$k$ expert: $w_{\text{top-}k} = \text{topk}(w, k)$
6: Set all other weights to zero: $w_i = 0$ for $i \neq \text{top-}k$
7: **if** training **then**
8:     **for each** $i \in \mathcal{E}$ **do**
9:         $y_i = \text{SP-SSM}(x_1, s_i) \odot \sigma(x_2)$
10:     **end for**
11:     Compute final output: $y = \sum_{i=1}^{n} w_i \cdot y_i$
12: **else**
13:     Compute final output: $y = w_{\text{top-}k} \cdot y_{\text{top-}k}$
14: **end if**
15: **Output:** Final output $y$

---

## C.1 COMPARISON TO LIGHTWEIGHT SR MDOELS (×3).

In Table 6, 7, we present the performance of our SP-MoMamba-T and SP-MoMamba-B model for ×3 upscaling, extending the results from Table 1, 2 in the main text. Our SP-MoMamba-T and SP-MoMamba-B can achieve competitive performance with lower parameter and computational complexity.

| Scale | Model | Params (M)↓ | GMACs↓ | Set5 PSNR↑ | Set5 SSIM↑ | Set14 PSNR↑ | Set14 SSIM↑ | BSD100 PSNR↑ | BSD100 SSIM↑ | Urban100 PSNR↑ | Urban100 SSIM↑ | Manga109 PSNR↑ | Manga109 SSIM↑ |
|---|---|---|---|---|---|---|---|---|---|---|---|---|---|
| ×3 | CARN-M | 415K | 46 | 33.99 | 0.9236 | 30.08 | 0.8367 | 28.91 | 0.8000 | 27.55 | 0.8385 | — | — |
| | PAN | 261K | 39 | 34.40 | 0.9271 | 30.36 | 0.8423 | 29.11 | 0.8050 | 28.11 | 0.8511 | 33.61 | 0.9448 |
| | DRSAN | 410K | 43 | 34.41 | 0.9272 | 30.27 | 0.8413 | 29.08 | 0.8056 | 28.19 | 0.8529 | — | — |
| | SAFMN | 233K | 23 | 34.34 | 0.9270 | 30.33 | 0.8418 | 29.08 | 0.8048 | 27.95 | 0.8474 | 33.52 | 0.9437 |
| | SeemoRe-T | 225K | 20 | 34.46 | 0.9276 | 30.44 | 0.8445 | 29.15 | 0.8063 | 28.27 | 0.8538 | 33.92 | 0.9460 |
| | SRConvNet | 387K | 33 | 34.40 | 0.9272 | 30.30 | 0.8416 | 29.07 | 0.8500 | 28.04 | 0.8474 | 33.56 | 0.9443 |
| | **SP-MoMamba-T (ours)** | 264K | 37 | **34.53** | **0.9284** | **30.50** | **0.8448** | **29.19** | **0.8080** | **28.43** | **0.8572** | **34.16** | **0.9504** |

Table 6: Comparison to efficient SR models. PSNR (dB ↑) and SSIM (↑) metrics are reported on the Y-channel. **Best** and second best performances are highlighted. GMACs (G) are computed by upscaling to a 1280 × 720 HR image.

| scale | Model | Params (M) | GMACs | Set5 PSNR | Set5 SSIM | Set14 PSNR | Set14 SSIM | BSD100 PSNR | BSD100 SSIM | Urban100 PSNR | Urban100 SSIM | Manga109 PSNR | Manga109 SSIM |
|---|---|---|---|---|---|---|---|---|---|---|---|---|---|
| ×3 | SwinIR-Light | 918K | 111 | 34.62 | 0.9289 | 30.54 | 0.8463 | 29.20 | 0.8082 | 28.66 | 0.8624 | 33.98 | 0.9478 |
| | SRFormer-Light | 861K | 105 | 34.67 | 0.9296 | 30.57 | 0.8469 | 29.26 | 0.8099 | 28.81 | 0.8655 | 34.19 | 0.9489 |
| | SPIN | 569K | 176 | 34.65 | 0.9293 | 30.57 | 0.8464 | 29.23 | 0.8089 | 28.71 | 0.8627 | 34.24 | 0.9489 |
| | MambaIR-light | 913K | 149 | 34.63 | 0.9288 | 30.54 | 0.8459 | 29.23 | 0.8084 | 28.70 | 0.8631 | 34.12 | 0.9479 |
| | SeemoRe-L | 959K | 87 | 34.70 | 0.9297 | 30.60 | 0.8469 | 29.29 | 0.8101 | 28.86 | 0.8653 | 34.53 | 0.9496 |
| | CRAFT | 744K | 88 | 34.71 | 0.9295 | 30.61 | 0.8469 | 29.24 | 0.8093 | 28.77 | 0.8635 | 34.29 | 0.9491 |
| | MambaIRv2-light | 781K | 127 | 34.71 | **0.9297** | **30.68** | **0.8483** | 29.26 | 0.8098 | **29.01** | **0.8689** | 34.41 | 0.9497 |
| | **SP-MoMamba-B (ours)** | 550K | 75 | **34.71** | **0.9297** | 30.65 | 0.8478 | **29.29** | **0.8104** | 28.84 | 0.8652 | **34.67** | **0.9530** |

Table 7: Comparison to lightweight SR models. PSNR (dB ↑) and SSIM (↑) metrics are reported on the Y-channel. **Best** and second best performances are highlighted. GMACs (G) are computed by upscaling to a 1280 × 720 HR image.

## C.2 EVALUATION ON REAL SR

We conduct experiments for Real SR (×4) using the Real-ESRGAN degradation model to evaluate SP-MoMamba-T against current efficient SOTA SR models such as SAFMN, SPIN, and SeemoRe-T, see Table 8. We report the popular NR-IQA metrics (NIQE and BRISQUE) on the commonly used real-world image collection provided by SeemoRe. As illustrated in the results, our method achieves the lowest scores (6.42 in NIQE and 44.69 in BRISQUE), indicating superior perceptual quality. Additionally, we conduct a cross-dataset evaluation using testsets with more realistic degradation of different severity levels (DIV2K-I and DIV2K-II). In these quantitative comparisons re-

| Method | NIQE↓ | BRISQUE↓ | DIV2K-I (PSNR↑) | DIV2K-II (PSNR↑) |
|---|---|---|---|---|
| Bicubic | 7.65 | 58.29 | 26.30 | 25.71 |
| SAFMN | 7.19 | 51.39 | 26.80 | 26.77 |
| SPIN | 6.84 | 58.27 | 26.93 | 26.86 |
| SeemoRe-T | 6.53 | 45.53 | 27.07 | 27.01 |
| SP-MoMamba-T | 6.42 | 44.69 | 27.13 | 27.09 |

Table 8: Real SR performance. NIQE and BRISQUE are reported on the real image collection provided by SeemoRe. DIV2K-I and DIV2K-II performance reported as PSNR.

| Method | LOLv1 | | LOLv2-Real | |
|---|---|---|---|---|
| | PSNR ↑ | SSIM ↑ | PSNR ↑ | SSIM ↑ |
| UHDFourLi et al. (2023a) | 22.89 | 0.8147 | 19.42 | 0.7896 |
| RetinexformerCai et al. (2023) | 22.71 | 0.8177 | 22.79 | 0.8397 |
| DMFourLLIE Zhang et al. (2024) | 22.98 | 0.8273 | 22.71 | 0.8583 |
| UHDFormer Wang et al. (2024a) | 22.88 | 0.8370 | 19.71 | 0.832 |
| RetinexMamba Bai et al. (2024) | 23.15 | 0.8210 | 21.73 | 0.829 |
| MambaLLIE Weng et al. (2024) | 22.80 | 0.8315 | 21.85 | 0.8276 |
| CWNet Zhang et al. (2025b) | 23.60 | 0.8496 | 23.31 | 0.8641 |
| CIDNet Yan et al. (2025) | 23.81 | **0.8574** | **23.43** | 0.8622 |
| SP-MoMamba-T | 23.25 | 0.8570 | 24.23 | 0.8732 |

Table 9: Analysis of SP-MoMamba generalization. We provide a performance comparison of SP-MoMamba on low-light image enhancement task.

ported as PSNR, SP-MoMamba-T consistently outperforms the competing methods, demonstrating its robustness in handling complex real-world degradation.

## C.3 EVALUATION ON LOW-LIGHT IMAGE ENHANCEMENT

As shown in Table 9, we assess the generalization capability of our proposed method by extending its application to the low-light image enhancement task. We compare SP-MoMamba-T against comprehensive state-of-the-art approaches, including Transformer-based (e.g., Retinexformer) and CNN-based (e.g., CIDNet) methods on the LOLv1 and LOLv2-Real datasets. Notably, our model yields significant enhancements, particularly on the more challenging LOLv2-Real benchmark. Specifically, SP-MoMamba-T outperforms the second-best method, CIDNet, achieving pronounced improvements of 0.80 dB in PSNR and 0.011 in SSIM. This phenomenon can be attributed to the robust global modeling capability of our superpixel-based architecture, which effectively restores illumination while preserving structural details. Overall, these advancements culminate in superior performance metrics, underscoring the flexibility and strong generalization potential of our design beyond the super-resolution task.

| Method | Time (s) | GPU Memory | PSNR |
|---|---|---|---|
| SwinIR-Light | 0.631 | 6802.7 | 30.91 |
| SRFormer-Light | 0.734 | 7319.4 | 31.17 |
| SPIN | 0.654 | 7083 | 30.98 |
| Mambairv2-Light | 0.833 | 13652.3 | 31.24 |
| CATANet | 0.745 | 21962 | 31.31 |
| SeemoRe-L | 0.145 | 10464.7 | 31.48 |
| SP-MoMamba-T | **0.084** | **4258.1** | 31.01 |
| SP-MoMamba-B | 0.236 | 6572.9 | **31.51** |

Table 10: Comparison between performance vs Inference times and GPU Memory on Manga109 ×4 dataset. Inference times and GPU Memory are calculated on 720p HR image.

## C.4 DETAILED COMPARISON OF MODEL COMPLEXITY

We provide a detailed comparison of memory usage and runtime as shown in Table 10. From the table, it can be seen that compared to the current best method, our SP-MoMamba-T has lowest inference time and GPU memory, and achieves more competitive performance.

# D MORE ABLATIONS

## D.1 ABLATION FOR LMA.

As a core component of the LSME module, the Local Mixed Attention (LMA) employs diverse strategies for focusing on local information, thereby effectively enhancing localized feature representation. Table 11 demonstrates that window multi-head self-attention (Swin MHSA) Liang et al. (2021), owing to its efficient window-based self-attention mechanism, achieves superior performance compared to the channel attention (CA) mechanism Zhang et al. (2018). However, Swin MHSA is restricted to spatially capturing local information, overlooking similarity relationships between different channels. Consequently, integrating CA with Swin MHSA can significantly enhance local features, leading to finer textures and improved performance. As demonstrated in Table 11, with the incorporation of CA and Swin MHSA, the model achieves optimal performance.

| Method | Params | GMACs | BSD100 | Urban100 |
|--------|--------|-------|--------|----------|
| Swin MHSA | 224K | 20 | 27.67 | 26.28 |
| CA | 240K | 13 | 27.65 | 25.22 |
| LMA | 271K | 22 | **27.69** | **26.40** |

Table 11: Ablation on the Local Mixed Attention (LMA) mechanism. We show results for ×4 upscaling.

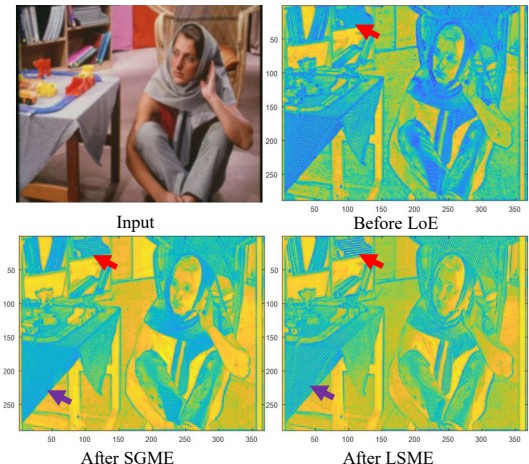

Figure 8: Visualizations of feature maps before and after applying the proposed SGME and LSME modules, demonstrating enhanced activation sharpness through SGME and refined representations via LSME. (Zoom in for the best view)

## D.2 ABLATION FOR SUPERPIXEL SAMPLING

We quantitatively compared our approach against hard clustering (SLIC ) and learnable token pooling (e.g., STA ). As shown in Table 12, our SSN-based strategy exhibits superior convergence and

| Method | Params | GMACs | BSD100 | Urban100 |
|--------|--------|-------|--------|----------|
| SP-MoMamba-T | | | | |
| w SLIC Achanta et al. (2012) | 271K | 26 | 27.51 | 26.03 |
| w STA Huang et al. (2022) | 460K | 38 | 27.72 | 26.49 |
| w SSN (Ours) | 271K | 22 | 27.69 | 26.30 |

Table 12: Superpixel sampling method. We compared the performance impact of different superpixel sampling method separately.

| Method | aggregation method | Urban100 | Manga109 |
|---|---|---|---|
| SP-MoMamba-T | w/o gate mechanism | 32.11 | 38.96 |
| | w add | 32.15 | 38.99 |
| | w gate mechanism | 32.22 | 39.01 |

Table 13: Analysis of gating Mechanism on SP-SSM. We provide further insights in the design decisions of our SP-MoMamba-T framework for ×2 upscaling.

numerical stability compared to token pooling methods, which often suffer from token collapse. This confirms that our differentiable clustering provides a more robust foundation for the pipeline.

### D.3 THE IMPACT OF GATE CONTROL MECHANISM ON PERFORMANCE IN SP-SSM

As evidenced in Table 13, removing the gating module or replacing it with element-wise addition results in performance degradation. This quantitatively confirms that the gating mechanism is indispensable for the model's reconstruction capability.

### D.4 FEATURE VISUALIZATION.

To substantiate the importance of the proposed SGME and LSME modules, we analyzed the feature maps before and after their integration into the Layer of Experts (LoEs), as illustrated in Figure 11. This analysis vividly highlights the strengths of employing MSS-MoE within the SGME module for global information extraction and the advantages of utilizing Swin MHSA combined with CA in the LSME module for local information refinement. Specifically, as indicated by the red arrows, structural global textures are markedly enhanced in the SGME module and subsequently refined further in the LSME module. Notably, as shown by the purple arrows, textures lost during the SGME filtering process are effectively recovered and complemented in the LSME stage, thereby synergistically enhancing the feature representation.

| Method | experts | Num superpixels | GMACs | BSD100 | Urban100 |
|---|---|---|---|---|---|
| | 3 | **[64,64,64]** | 22 | **27.69** | **26.40** |
| SP-MoMmaba-T | 3 | [16,32,64] | **21** | 27.65 | 26.28 |
| | 3 | [32,64,128] | 39 | 27.66 | 26.30 |

Table 14: Analysis of the impact of the number of superpixels on the performance of MSS-MoE.

### D.5 NUMBER OF SUPERPIXEL ON MSS-MOE

Since superpixels represent the most relevant pixels within the same semantic region, a higher number of superpixels leads to finer semantic segmentation. To further investigate the impact of superpixel quantity on the performance of MSS-MoE, we conducted a series of ablation experiments, as presented in Table 14. The results indicate that increasing the number of superpixels does not necessarily yield better performance; it has minimal impact on network accuracy but noticeably affects computational efficiency. This stems from the fact that a greater number of superpixels merely subdivides identical semantic regions without capturing semantic information across different scales. Consequently, incorporating multi-scale superpixels proves more effective in enhancing model performance.

| Method | L1 | FFT | BSD100 | Urban100 |
|---|---|---|---|---|
| | 1.0 | 0.0 | 27.65 | 26.26 |
| SP-MoMmaba-T | **1.0** | **0.1** | **27.69** | **26.40** |
| | 1.0 | 0.2 | 27.68 | 26.32 |

Table 15: Loss function. SP-MoMamba was trained on DIV2K and Flickr2K. We report PSNR (dB ↑) on the Y-Channel for ×4 upscaling.

### D.6 Loss Function

We use FFT loss and L1 loss to jointly optimize the network. The specific expression is as follows:

$$L_{total} = L_1 + w * L_{FFT} \tag{16}$$
$$L_1 = ||I_{gt} - I_{SR}||_1 \tag{17}$$
$$L_{FFT} = ||FFT(I_{gt}) - FFT(I_{SR})||_1 \tag{18}$$

where the $FFT(\cdot)$ denotes the Fourier Transformation. The $I_{gt}$ and $I_{SR}$ represent the ground-truth image and super-resolution image. To verify the effectiveness of the loss function, we designed a group of ablation experiments, as shown in Table 15. As can be seen from the table, compared with using only L1 loss, the added FFT loss can effectively add constraints in the frequency domain to the model, so as not to over smooth the texture and make the performance better.

## E Visual Results

In Figure 9, we provide additional visual comparisons for the BSD100 benchmark, and in Figure 10 for the Urban100 benchmark ($\times 4$). Our SP-MoMamba framework consistently delivers visually appealing results, even when applied to intricate architectural structures. For example, as demonstrated by img021 in Figure 9, our model significantly outperforms other methods in reconstructing patterns with greater fidelity. Moreover, the reconstruction of img096 in Figure 10 exhibits reduced blur and sharper edges, thereby improving overall visual clarity.

We also provide additional the LAM results and Diffusion Index (DI) in Figure 11. It validates semantic preservation in natural scenes containing discrete, similarly textured objects. While MambaIR and MambaIRv2 display localized activation, failing to link spatially separated instances, SP-MoMamba demonstrates broader coverage by effectively leveraging information from neighboring objects. This superior capture of non-local correlations is quantitatively confirmed by the highest DI score of 24.36 (vs. 19.28/23.91), proving our method overcomes 1D scanning limitations to maintain semantic integrity.

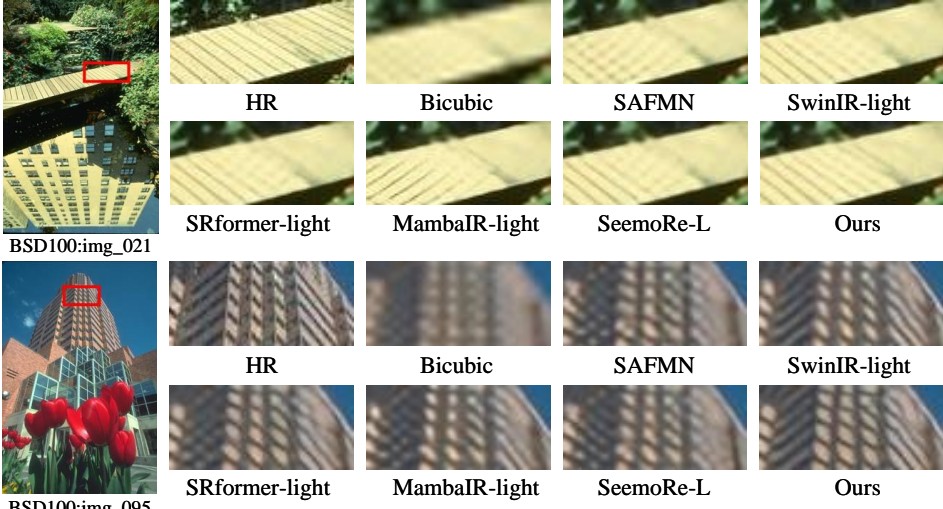

Figure 9: Visual comparison of SP-MoMamba with state-of-the-art methods on challenging cases for $\times 4$ SR from the BSD100 benchmark

## F Failure Cases and Limitation

Despite the impressive balance between efficiency and performance achieved by our proposed SP-MoMamba, we acknowledge certain limitations inherent to our design. As illustrated in the failure case in the Figure 12, the model encounters difficulties when reconstructing scenes containing

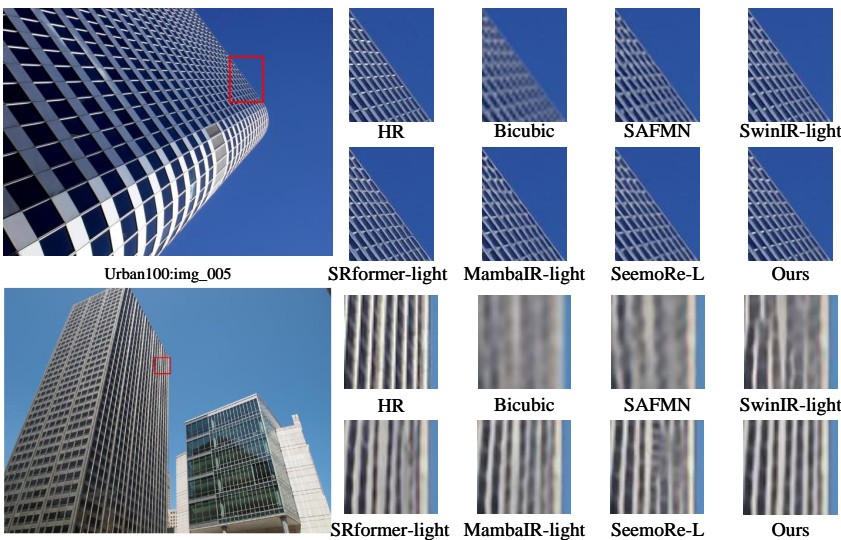

Figure 10: Visual comparison of SP-MoMamba with state-of-the-art methods on challenging cases for ×4 SR from the Urban100 benchmark

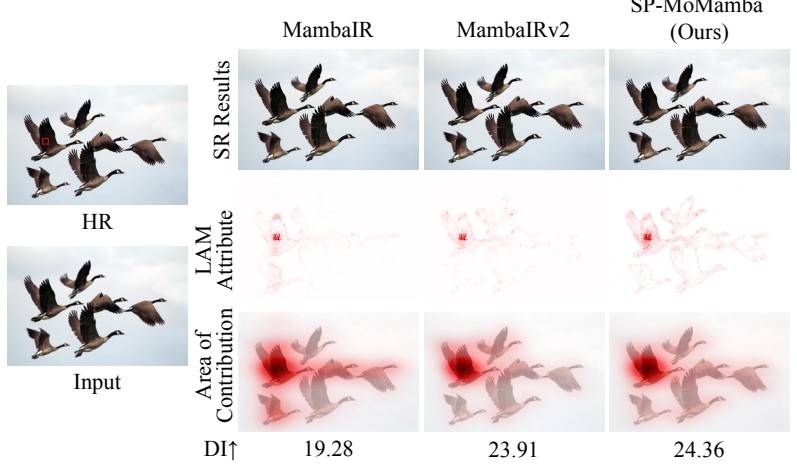

Figure 11: Analysis semantic preservation capability of different methods. The larger the Diffusion Index (DI), the more semantically similar pixels are involved in restoring the corresponding region. (Zoom in for the best view)

extremely dense, high-frequency repetitive patterns under severe degradation. Notably, while our method recovers sharper edges compared to competing models like Mambairv2-light, it inadvertently introduces slight geometric distortions and aliasing artifacts within the fine grid structures. This phenomenon can be attributed to the dependency on superpixel-guided priors; our core motivation employs superpixels to maintain semantic structure and mitigate the semantic destruction typically caused by traditional Mamba scanning. However, when dealing with extremely blurry or noisy low-resolution inputs, superpixel algorithms may struggle to generate semantic regions with precise edge alignment. In fact, this geometric inaccuracy serves to mislead the subsequent State Space Models (SSM), thereby affecting the fidelity of the final reconstruction results. Consequently, while the global semantic layout is preserved, the local high-frequency precision is compromised in these extreme scenarios.

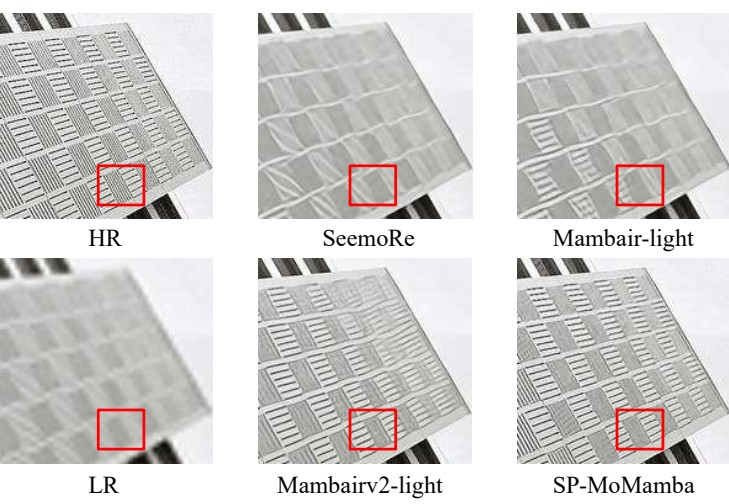

Figure 12: Visual comparison of SP-MoMamba with state-of-the-art methods on challenging cases for ×4 SR from the Urban100 benchmark

