# OpenReview forum: "SP-MoMamba: Superpixel-driven Mixture of State Space Experts for Efficient Image Super-Resolution"
_ICLR.cc/2026/Conference — Submitted to ICLR 2026_

### Official Review · Reviewer_L23R · 2025-10-18

**Soundness:** 3
**Presentation:** 2
**Contribution:** 2
**Rating:** 6
**Confidence:** 4

**Summary:**

This paper proposes SP-MoMamba for the super-resolution (SR) task, which combines super-pixels and mixture of experts with state space models (SSM). The potential contributions are:
(1) an SP-SSM that combines superpixel sampling and SSM through a gating mechanism,
(2) an MSS-MoE that uses a routing mechanism to dynamically select features from multiple SP-SSM experts.
The method is evaluated on five standard SR benchmarks and demonstrates a compelling performance-efficiency trade-off: achieving competitive results with lower computational overheads.

**Strengths:**

**Clarity and Reproducibility:** The paper is generally well-written and easy to follow. The inclusion of pseudo-code and the source code significantly strengthens the paper's credibility and reproducibility.

**Technical Interest:** The idea of exploring structured representations (super-pixels) with SSMs is interesting.

**Empirical Performance:** The results are impressive. The method achieves state-of-the-art or highly competitive performance across multiple datasets, with lower computational costs, including GMACs, inference times, GPU memory usage, etc. Figures 5, 7, and 8 provide compelling visual results.

**Weaknesses:**

**Weak Motivation:**
The paper positions itself as the first work to combine super-pixels with SSMs for SR. However, the motivation for this specific combination is not sufficiently developed. Super-pixel has been explored with transformers for SR in (Zhang et al. 2023). Considering SSMs are a popular substitute for transformers, and there are already Mamba-based SR methods (e.g., (Gut et al., 2024; Qiao et al. 2024)), combining the super-pixel with SSMs instead of transformers is not significant.

A strong narrative should answer: What specific limitations of existing SSM-based SR methods (e.g., high frequency details and/or semantic inconsistency) can be effectively addressed by superpixels? Meanwhile, what limitations of super-pixel-based methods are addressed by the SSM? What are the challenges of combining these two? The current paper does not provide a new insight, making the core idea feel more like a clever stacking of existing techniques than a novel solution to a well-defined problem.



**Unclear Novelty and Justification of Architectural Complexity:**
SP-SSM and MSS-MoE are claimed as two major contributions/novelties. However, these two modules are complex compositions of established techniques. The SP-SSM is a combination of super-pixel sampling (Jampani et al., 2018), SSM, and Gumbel-Softmax (Jang et al., 2016) through a gating mechanism. The MSS-MoE is mainly a combination of SP-SSM and the routing mechanism. The paper will need to address the following:

(1) Explain the necessities of the designs. For example, why is this gating mechanism in SP-SSM the best choice for combining super-pixels and SSM? What is the insight behind it?

(2) Demonstrate their effectiveness over simple baselines. For example, how does the MSS-MoE perform against a simple baseline of a non-dynamic ensemble of experts? Does the routing mechanism play a key role, or does the performance benefit from a large aggregation model?

(3) Quantify individual contributions. The ablation study (Table 3) is incomplete. The contributions of many design choices,  such as the GatedFFN, the residual scaling parameters, and the specific formulations of Eqs. (1, 5-7) are not reflected.

The core issue is that this method feels over-engineered, with many components whose individual necessity and novelty are not rigorously established. This buries the true potential research contribution under a layer of engineering complexity. The strong results may be attributed to the overall capacity and designs, but the scientific insight remains unclear.


**Justification:** This paper proposes a new method for super-resolution, whose efficiency and performance results are compelling. The major concerns are about insufficient articulation of its conceptual novelty and justification of the necessity of its complex design through rigorous ablation. The contribution seems more from a strong engineering effort rather than a clear scientific insight. The authors will need to address these concerns during the rebuttal.

**Questions:**

1. Beyond the empirical results, what is the core scientific insight from combining superpixels with SSMs? Can the authors explain why this combination is particularly powerful for SR (but not other dense prediction tasks)?

2. Could the authors provide ablations comparing SP-SSM and MSS-MoE to their corresponding strong/simpler baselines? For example, SP-SSM vs. SP-SSM (w/o gating), and MSS-MoE vs. a direct fusion of experts.

3. Please consolidate the ablation studies into a single table in the main paper (e.g., expanding Table 3 to include ablations mentioned above and those from Table 9). This is essential for the readers to understand the contribution of each part of this method.

4. The method appears to be a general-purpose dense prediction architecture. Have the authors validated it on other tasks (e.g., segmentation, enhancement)?

5. Please provide a discussion on any failure cases and analyze the intrinsic limitations of this method in the context of SR.

---

> ### Author Response · Authors · 2025-11-26
> **Response to Reviewer L23R (1/3)**
>
> Thank you for your constructive comments and suggestions. We have revised our paper according to your comments. We respond to your questions below and would appreciate it if you could let us know if our response addresses your concerns.
> > **Q1:** The paper positions itself as the first work to combine super-pixels with SSMs for SR. However, the motivation for this specific combination is not sufficiently developed. Super-pixel has been explored with transformers for SR in (Zhang et al. 2023). Considering SSMs are a popular substitute for transformers, and there are already Mamba-based SR methods (e.g., (Gut et al., 2024; Qiao et al. 2024)), combining the super-pixel with SSMs instead of transformers is not significant.
>
> **A1:** We respectfully clarify that SP-MoMamba is not a trivial combination but a targeted architectural solution. The synergy between Superpixels and SSMs addresses specific limitations:
>
> 1. **Why Mamba Needs Superpixels (Solving Semantic Disruption):** Unlike Transformers, Mamba-based methods rely on 1D scanning, which inherently severs local spatial connections. Superpixels group pixels by semantic similarity before serialization. This acts as a critical remedy to the **semantic disruption** caused by SSM scanning—a benefit unique to the SSM architecture. For details, please refer to Response reviewer nJJj's A1 answer and you can see Figure 11 of the manuscript.
>
> 2. **Why Superpixels Need SSMs (Efficiency at Scale):** Previous Superpixel-Transformer works (e.g., SPIN) suffer from quadratic complexity ($ O(N^2)$ ). By leveraging SSM's linear complexity ($ O(N)$ ), we enable efficient global modeling among irregular superpixels. This allows our method to scale significantly better on high-resolution inputs compared to Transformer baselines.
>
> 3. **Our Technical Innovation:** Combining irregular superpixels with ordered SSM scanning presents unique challenges, such as information imbalance and loss of fine-grained detail. We addressed these via our novel **MSS-MoE** architecture, which introduces **dynamic routing** to handle varying information density and Local Spatial Modulation Experts (**LSME**) to compensate for local detail loss.
>
> In summary, this design is a mutually beneficial strategy: Superpixels fix Mamba's semantic breaking, while SSMs solve the efficiency bottleneck of Superpixel-Transformers.
>
> > **Q2:** Explain the necessities of the designs. For example, why is this gating mechanism in SP-SSM the best choice for combining super-pixels and SSM? What is the insight behind it.
>
> **A2:** We justify the design of the gating mechanism in SP-SSM from three key perspectives:
>
> 1. **Design Insight (Resolving Granularity Mismatch):** Superpixels represent coarse semantic units, whereas SR demands fine pixel-level details. Direct fusion often leads to artifacts. The gating mechanism acts as a **semantic valve**, selectively injecting global context from the SSM only where beneficial, thereby preventing the over-smoothing of local textures preserved in the residual path.
>
> 2. **Technical Superiority:** Compared to simple add and concat operation, our gating leverages the differentiable mask (from Gumbel-Softmax) to precisely project inter-superpixel relationships back to the pixel domain. This ensures that global attention is spatially aligned with specific semantic regions, effectively balancing global coherence with local fidelity.
>
> 3. **Empirical Verification:** As evidenced in Table R2, removing the gating module or replacing it with element-wise addition results in performance degradation. This quantitatively confirms that the gating mechanism is indispensable for the model's reconstruction capability. We have revised the paper and put these details in Appendix D.3.
>
> **Table R2. Analysis of gating Mechanism on SP-SSM.**
>
> | Method         | aggregation method | Urban100 | Manga109 |
> | :------------- | :----------------- | :------: | :------: |
> | SP-MoMamba-T   | w/o gate mechanism |  32.11   |  38.96   |
> | SP-MoMamba-T   | w add              |  32.15   |  38.99   |
> | SP-MoMamba-T   | w gate mechanism   |  32.22   |  39.01   |

---

> ### Author Response · Authors · 2025-11-26
> **Response to Reviewer L23R (2/3)**
>
> > **Q3:** Demonstrate their effectiveness over simple baselines. For example, how does the MSS-MoE perform against a simple baseline of a non-dynamic ensemble of experts? Does the routing mechanism play a key role, or does the performance benefit from a large aggregation model?
>
> **A3:** We sincerely appreciate your valuable suggestion. Accordingly, we have conducted ablation studies on the MSS-MoE to verify the necessity of the Mixture-of-Experts mechanism, as presented in Table 4 (a) of the manuscript. When $ \\#\mathcal{E}$  in the table is 1, it represents a simple baseline of a non-dynamic ensemble of experts. We also analyzed the impact of aggregating different numbers of experts $ k$  on performance in the appendix, as shown in Table 4 (b) of the manuscript. These results indicate that the routing mechanism effectively leverages complementary semantic information across different scales. By dynamically selecting the optimal experts via the Router, the model significantly enhances its capability for detailed reconstruction. Crucially, this design ensures that computational complexity remains independent of the total number of experts, thereby achieving an optimal balance between high performance and minimal computational overhead. We have revised the paper and put these details in Section 5.3 of the manuscript.
>
> > **Q4:**  Quantify individual contributions. The ablation study (Table 3) is incomplete. The contributions of many design choices, such as the GatedFFN, the residual scaling parameters, and the specific formulations of Eqs. (1, 5-7) are not reflected.
>
> **A4:** Following your suggestion, we have further refined the ablation study presented in Table R4. From the table, it can be seen that the module we proposed can effectively improve the performance of the model without increasing a lot of computational complexity. We have revised the manuscript accordingly and incorporated these details into the **Ablation Study** section.
>
> **Table R4. Ablation on key components of SP-MoMamba.**
>
> | Method         | SSM | SPS | MoE | LMA | GatedFFN | Params | GMACs | BSD100 | Urban100 |
> | :------------- | :-: | :-: | :-: | :-: | :------: | :----: | :---: | :----: | :------: |
> | Baseline       | -   | -   | -   | -   | -        | 189K   | 8.4   | 27.42  | 25.55    |
> | SP-MoMamba-T   | ✓   | -   | -   | -   | -        | 230K   | 18.7  | 27.48  | 25.64    |
> | SP-MoMamba-T   | ✓   | ✓   | -   | -   | -        | 197K   | 16.7  | 27.54  | 25.86    |
> | SP-MoMamba-T   | ✓   | ✓   | ✓   | -   | -        | 242K   | 16.9  | 27.59  | 26.07    |
> | SP-MoMamba-T   | -   | -   | -   | ✓   | ✓        | 211K   | 16.6  | 27.61  | 26.14    |
> | SP-MoMamba-T   | ✓   | ✓   | ✓   | -   | ✓        | 261K   | 19.8  | 27.63  | 26.14    |
> | SP-MoMamba-T   | ✓   | ✓   | ✓   | ✓   | ✓        | 272K   | 22    | 27.69  | 26.40    |

---

> ### Author Response · Authors · 2025-11-26
> **Response to Reviewer L23R (3/3)**
>
> > **Q5:** Beyond the empirical results, what is the core scientific insight from combining superpixels with SSMs? Can the authors explain why this combination is particularly powerful for SR (but not other dense prediction tasks)?
>
> **A5:** We articulate how our design reconstructs the input paradigm to resolve inherent limitations in Mamba-based SR from three perspectives:
>
> 1. **Fundamental Insight (Mitigating Semantic Disruption):** Existing vision SSMs rely on 2D Selective Scan (SS2D), which forces images into 1D sequences, inevitably severing spatial connections and causing artifacts. Our core insight is to reconstruct the scanning methodology to operate on superpixels rather than pixels, preserving 2D spatial coherence even within a 1D stream. For details, please refer to Response reviewer nJJj's A1 answer and you can see Figure 11 of the manuscript.
>
> 2. **Methodological Innovation (Superpixels as Semantic Units):** By using superpixels—which naturally cluster semantic similar pixels—as the fundamental scanning units, we effectively resolve the "semantic interruption" inherent in Mamba. Simultaneously, this compresses the feature space, allowing the SSM to maintain high semantic accuracy while preserving its linear computational efficiency.
>
> 3. **Architectural Synergy (Multi-Scale Collaboration via MoE):** Beyond the scanning mechanism, we employ a Mixture-of-Experts (MoE) framework. This design optimally integrates the semantic preservation of superpixels with the global modeling of SSMs, achieving a synergistic balance between semantic coherence, detail reconstruction, and efficiency.
>
> In addition, we have extended our method to the field of low light image enhancement, which shows that our method can also achieve competitive results, as shown Table R5.
>
> **Table R5. Comparison of SP-MoMamba on low-light image enhancement task.**
>
> | Methods                     | LOLv1 PSNR ↑ | LOLv1 SSIM ↑ | LOLv2-Real PSNR ↑ | LOLv2-Real SSIM ↑ |
> | :-------------------------- | :----------: | :----------: | :---------------: | :---------------: |
> | UHDFour (Li et al. 2023)    | 22.89        | 0.8147       | 19.42             | 0.7896            |
> | Retinexformer (Cai et al. 2023) | 22.71        | 0.8177       | 22.79             | 0.8397            |
> | DMFourLLIE (Zhang et al. 2024) | 22.98        | 0.8273       | 22.71             | 0.8583            |
> | UHDFormer (Wang et al. 2024) | 22.88        | 0.8370       | 19.71             | 0.8320            |
> | RetinexMamba (Bai, Yin, and He 2024) | 23.15        | 0.8210       | 21.73             | 0.8290            |
> | MambaLLIE (Weng et al. 2025) | 22.80        | 0.8315       | 21.85             | 0.8276            |
> | CWNet (Zhang et al. 2025)   | 23.60        | 0.8496       | 23.31             | 0.8641            |
> | CIDNet (Yan et al. 2025)    | 23.81        | **0.8574**   | 23.43             | 0.8622            |
> | URetienxNet++ (Yan et al. 2025) | **23.83**        | 0.8390       | 21.97             | 0.8360            |
> | SP-MoMamba-T                | 23.25        | 0.8570       | **24.23**         | **0.8732**        |
>
> > **Q6:** The method appears to be a general-purpose dense prediction architecture. Have the authors validated it on other tasks (e.g., segmentation, enhancement)?
>
> **A6:** We sincerely appreciate your valuable suggestion. Following your recommendation, we have explored the performance of our model on image enhancement tasks, as presented in Table R5. The results demonstrate that our method achieves competitive performance in this domain, thereby further validating the architectural advantages of our approach in maintaining semantic continuity and balancing global versus local detail precision. In future work, we plan to extend our investigation to evaluate the model's performance across a broader range of diverse vision tasks. We have revised the paper and put these details in Appendix C.3.
>
> > **Q7:** Please provide a discussion on any failure cases and analyze the intrinsic limitations of this method in the context of SR.
>
> **A7:** Despite SP-MoMamba's impressive balance between efficiency and performance in lightweight super-resolution (SR), we acknowledge a limitation inherent to its design. Our approach relies on superpixels to preserve semantic structure and mitigate the semantic destruction typically caused by traditional Mamba scanning. However, when dealing with extremely blurry or noisy low-resolution inputs, superpixel algorithms may struggle to generate semantic regions with precise edge alignment. This geometric inaccuracy can mislead the subsequent State Space Models (SSM), thereby compromising the fidelity of the final reconstruction. We have revised the paper and put specific examples (see Figure 12) and details in Appendix F.

---

> > ### Comment · Reviewer_L23R · 2025-11-27
> >
> > Thank you for your responses and the new results. My previous concerns/comments have been addressed.
> >
> > A minor suggestion is to check and consider using another dataset instead of LOLv1 for the evaluation of low-light image enhancement. If I remember correctly, there are scene overlaps between the training and test sets, which may make the comparison less fair.

---

### Official Review · Reviewer_iSvu · 2025-10-30

**Soundness:** 3
**Presentation:** 3
**Contribution:** 3
**Rating:** 6
**Confidence:** 5

**Summary:**

This paper proposes SP-MoMamba, an image super-resolution method that integrates Superpixel and State Space Model (SSM), including the Superpixel-driven State Space Model (SP-SSM) and Multi-Scale Superpixel Mixture of State Space Experts (MSS-MoE) modules. The method enhances modeling efficiency while preserving semantic structure integrity, and achieves superior performance over existing lightweight methods on multiple mainstream datasets.

**Strengths:**

1、English expression in this paper is of high quality.
2、The motivation section reasonably identifies the issue of texture and semantic information disruption in existing Mamba-based methods, while the proposed solution demonstrates considerable inspiration and novelty.
3、objective metrics (PSNR, SSIM), number of parameters, and GMACs all surpass those of existing methods, demonstrating excellent performance.
4、Experiments exhibit a high level of completeness, with thorough comparative analyses and relatively comprehensive ablation studies.

**Weaknesses:**

1、Potential Lack of Novelty:
SPIN [1] has already employed superpixels + attention mechanisms for super-resolution (SR). This paper merely replaces attention with an SSM (State Space Model) without sufficiently justifying the unique advantages of SSMs in superpixel modeling. The authors should emphasize why the combination of Mamba and superpixels is particularly reasonable. Relying solely on experimental results for justification lacks persuasiveness.
2、There are some textual errors. For example, in Line 90: "Technically, our SP-MoMamba is composed of stacked Layer of Experts" – "Layer" should be "Layers".
Table1 (Line 325 ), PSNR of CARN-M on set14 is "33.26" not :33..26"

Reference: [1]Zhang, Aiping, et al. "Lightweight image super-resolution with superpixel token interaction." Proceedings of the IEEE/CVF international conference on computer vision. 2023.

**Questions:**

Please kindly ask the authors to focus their rebuttal on addressing Weakness 1.

---

> ### Author Response · Authors · 2025-11-26
> **Response to Reviewer iSvu**
>
> Thank you for your constructive comments and suggestions. We have revised our paper according to your comments. We respond to your questions below and would appreciate it if you could let us know if our response addresses your concerns.
> > **Q1:** Potential Lack of Novelty: SPIN [1] has already employed superpixels + attention mechanisms for super-resolution (SR). This paper merely replaces attention with an SSM (State Space Model) without sufficiently justifying the unique advantages of SSMs in superpixel modeling. The authors should emphasize why the combination of Mamba and superpixels is particularly reasonable. Relying solely on experimental results for justification lacks persuasiveness.
>
> **A1:** We clarify that our work differs fundamentally from SPIN [1]. We do not merely replace Attention with SSM; rather, we identify a unique **"Problem-Solution Fit"** between Superpixels and SSMs that addresses challenges specific to Mamba architectures:
>
> 1. **Theoretical Compatibility (Solving SSM-Specific "Semantic Disruption"):** SPIN uses superpixels primarily to reduce token count for Attention, which is permutation-invariant. In contrast, Mamba is strictly order-dependent, and standard raster scanning destroys 2D spatial consistency. We use superpixels as **scanning units** to preserve semantic continuity during serialization. This addresses the **Semantic Disruption** flaw inherent to SSMs—a critical structural issue that Attention-based methods like SPIN do not encounter. For details, please refer to Response reviewer nJJj's A1 answer and you can see Figure 11 of the manuscript.
>
> 2. **Computational Superiority (Linear $ O(M)$  vs. Quadratic $ O(M^2)$ ):** While SPIN reduces resolution, its inter-superpixel attention modeling remains quadratic ($ O(M^2)$ ), limiting scalability on high-resolution inputs. SP-MoMamba leverages SSM's linear complexity ($ O(M)$ ) to achieve efficient global modeling. This makes our method significantly more scalable and suitable for resource-constrained SR tasks compared to SPIN.
>
> 3. **Architectural Uniqueness (MSS-MoE):** Simply swapping Attention for SSM on irregular superpixels risks detail loss due to state compression. Unlike SPIN, we designed the Multi-Scale Superpixel Mixture of State Space Experts (MSS-MoE). We introduce **dynamic routing** and Local Spatial Modulation Experts (**LSME**) to specifically compensate for the information loss caused by the Superpixel-SSM combination, ensuring a tailored synergy absent in SPIN.
>
> In summary, integrating Mamba and Superpixels is not a trivial substitution but a **necessary reconstruction** to fix SSM scanning defects and unlock linear scalability, supported by our novel MoE architecture.
>
> [1] Zhang A, Ren W, Liu Y, et al. Lightweight image super-resolution with superpixel token interaction[C]//Proceedings of the IEEE/CVF international conference on computer vision. 2023: 12728-12737.
>
> > **Q2:** There are some textual errors. For example, in Line 90: "Technically, our SP-MoMamba is composed of stacked Layer of Experts" – "Layer" should be "Layers". Table1 (Line 325 ), PSNR of CARN-M on set14 is "33.26" not :33..26".
>
> **A2:**  We sincerely thank the reviewer for the meticulous review and for pointing out these critical textual and formatting errors. Following your suggestions, we have corrected all the mentioned issues and conducted a thorough proofreading of the revised manuscript to ensure it is free of such errors.

---

### Official Review · Reviewer_TnNR · 2025-10-31

**Soundness:** 3
**Presentation:** 2
**Contribution:** 3
**Rating:** 6
**Confidence:** 4

**Summary:**

This paper proposes SP-MoMamba, a superpixel-driven state space modeling framework that addresses a fundamental limitation of Mamba-based image restoration models—namely, the semantic distortion introduced by flattening 2D images into 1D scan sequences. To preserve spatial coherence, the authors introduce SP-SSM, which aggregates semantically homogeneous pixels into superpixels prior to state-space modeling, thereby maintaining regional semantic consistency.

The overall architecture is composed of stacked Layers of Experts (LoE), each consisting of a Semantic-Guided Mamba Expert (SGME) for global structure modeling followed by a Local Spatial Modulation Expert (LSME) for fine-grained texture reconstruction. Within SGME, the model incorporates a Multi-Scale Superpixel Mixture-of-Experts (MSS-MoE) module, which performs sparse routing across multi-scale SP-SSM experts. All experts participate during training, while only the Top-k experts are activated at inference, making the inference cost essentially independent of the total number of experts.

Overall, the paper’s contribution lies in embedding semantic priors into state-space modeling via “superpixel representation + multi-scale MoE + sparse routing,” enabling improved long-range structural consistency while preserving local detail, and delivering strong performance under constrained computational budgets.

**Strengths:**

This paper introduces a novel perspective for improving state-space models in image super-resolution by addressing the semantic disruption caused by 2D-to-1D scanning. The proposed superpixel-driven state-space formulation preserves regional semantic consistency and represents a creative integration of perceptual grouping principles with efficient sequential modeling. The use of multi-scale superpixel mixture-of-experts and sparse routing further strengthens the approach, enabling an effective balance between global structure modeling and fine-grained detail restoration. Empirical results across standard benchmarks demonstrate consistent gains over strong lightweight baselines, highlighting both the technical quality and practical significance of the contribution.

**Weaknesses:**

While the paper presents a compelling framework, several areas warrant further development to strengthen its contribution. First, the reliance on pre-defined superpixel scales introduces sensitivity to hyperparameter choices and may limit robustness across diverse visual domains; an adaptive mechanism or learning-based superpixel module would enhance generalization. Second, the evaluation focuses primarily on bicubic degradation, leaving open questions regarding performance under real-world or unknown degradation settings, where superpixel consistency may be more fragile. Third, although the MoE routing scheme is conceptually sound, deeper analysis of expert specialization (e.g., visualization of expert roles, load-balancing behavior[1][2][3]) would clarify the functional contribution of the mixture structure. Finally, while the paper situates its contributions within the Mamba-based SR literature, stronger comparison to recent frequency-aware and hybrid prior-guided SR models would further contextualize the novelty and demonstrate robustness across broader architectural trends.
[1]Dai, T., Wang, J., Guo, H., Li, J., Wang, J., & Zhu, Z. (2024, August). FreqFormer: Frequency-aware transformer for lightweight image super-resolution. In Proceedings of the International Joint Conference on Artificial Intelligence (pp. 731-739).
[2]Huang, F., Liu, H., Chen, L., Shen, Y., & Yu, M. (2025). Feature enhanced cascading attention network for lightweight image super-resolution. Scientific Reports, 15(1), 2051.
[3]Wang, Y., Liu, Y., Zhao, S., Li, J., & Zhang, L. (2024). CAMixerSR: Only Details Need More" Attention". In Proceedings of the IEEE/CVF Conference on Computer Vision and Pattern Recognition (pp. 25837-25846).

**Questions:**

1.Could you elaborate on how the superpixel clustering is integrated into the end-to-end training pipeline? For instance, is the clustering fully differentiable, and how is stability ensured when using Gumbel-Softmax for hard assignment? Additionally, were alternative soft segmentation strategies (e.g., learnable token pooling or differentiable k-means variants) evaluated, and how do they compare in terms of convergence and gradient behavior?

2.Can you provide more detail on the computational breakdown of SP-SSM during inference? While the paper emphasizes the overall efficiency, it would be helpful to quantify the latency contributions from superpixel generation, routing, and SSM inference separately. Also, how does the method scale on high-resolution inputs and resource-constrained devices, relative to Transformer-based SR models and MambaIR?

3.In your discussion on multi-scale expert routing, could you share empirical evidence that different experts specialize in distinct spatial scales or semantic structures? For example, visualizing expert activation patterns or analyzing usage frequency across datasets would help clarify whether the mixture-of-experts contributes meaningful functional diversity beyond parameter expansion.

4.To better substantiate the claimed advantages, could you add comparisons or re-train baselines against recent lightweight and frequency-aware SR models—e.g., FreqFormer [1], FECAN/FECA [2], and CAMixerSR [3]—under the same training protocol and evaluation settings (×2/×4, identical data and metrics)? Reporting PSNR/SSIM as well as perceptual metrics (LPIPS/DISTS) and latency would help position your method against the current state of the art.

[1]Dai, T., Wang, J., Guo, H., Li, J., Wang, J., & Zhu, Z. (2024, August). FreqFormer: Frequency-aware transformer for lightweight image super-resolution. In Proceedings of the International Joint Conference on Artificial Intelligence (pp. 731-739).
[2]Huang, F., Liu, H., Chen, L., Shen, Y., & Yu, M. (2025). Feature enhanced cascading attention network for lightweight image super-resolution. Scientific Reports, 15(1), 2051.
[3]Wang, Y., Liu, Y., Zhao, S., Li, J., & Zhang, L. (2024). CAMixerSR: Only Details Need More" Attention". In Proceedings of the IEEE/CVF Conference on Computer Vision and Pattern Recognition (pp. 25837-25846).

---

> ### Author Response · Authors · 2025-11-26
> **Response to Reviewer TnNR (1/3)**
>
> Thank you for reviewing our paper and for your valuable feedback. Below, we address your concerns point by point and we’ve revised our paper according to your suggestions. We would appreciate it if you could let us know whether your concerns are addressed by our response.
> > **Q1:** Could you elaborate on how the superpixel clustering is integrated into the end-to-end training pipeline? For instance, is the clustering fully differentiable, and how is stability ensured when using Gumbel-Softmax for hard assignment? Additionally, were alternative soft segmentation strategies (e.g., learnable token pooling or differentiable k-means variants) evaluated, and how do they compare in terms of convergence and gradient behavior?
>
> **A1:** Regarding the integration, differentiability, and stability of our clustering mechanism, we provide a detailed response from three perspectives:
>
> 1. **Technical Realization (Fully Differentiable End-to-End Integration)**:
> We employ an **SSN-based soft K-Means algorithm** to replace traditional hard clustering. Through $ T$  iterations (Eq. 9-10), we calculate the affinity matrix $ M_{sim}$  using entirely differentiable operations. This allows the network to dynamically learn the optimal superpixel directly from the SR loss, bypassing the limitations of fixed, non-differentiable segmentation methods.
>
> 2. **Stability Mechanism (Gumbel-Softmax for Robust Assignment):**
> To ensure stability, we utilize the **Gumbel-Softmax** operation to transform $ M_{sim}$  into a One-hot Mask $ M_{mask}$ . This acts as a bridge: enabling "hard" assignment during the forward pass for precise indexing, while maintaining a differentiable approximation for the backward pass. This guarantees smooth gradient flow from global features back to the pixel space, effectively preventing the training instability typically associated with discrete sampling.
>
> 3. **Comparative Verification (Superior Convergence vs. Alternatives):**
> We quantitatively compared our approach against hard clustering (SLIC [1]) and learnable token pooling (e.g., STA [2]). As shown in **Table R1**, our SSN-based strategy exhibits superior convergence and numerical stability compared to token pooling methods, which often suffer from token collapse. This confirms that our differentiable clustering provides a more robust foundation for the pipeline. We have revised the manuscript accordingly and included these details in Appendix D.2.
>
> [1] Achanta R, Shaji A, Smith K, et al. SLIC superpixels compared to state-of-the-art superpixel methods[J]. IEEE transactions on pattern analysis and machine intelligence, 2012, 34(11): 2274-2282.
> [2] Huang H, Zhou X, Cao J, et al. Vision transformer with super token sampling[J]. arXiv preprint arXiv:2211.11167, 2022.
>
> **Table R1.** **Ablation on superpixel sampling method.**
>
> | Method         | Params | GMACs | BSD100 | Urban100 |
> | :------------- | :----: | :---: | :----: | :------: |
> | SP-MoMamba-T   |        |       |        |          |
> | w SLIC         | 271K   | 26    | 27.51  |  26.03   |
> | w STA          | 460K   | 38    | 27.72  |  26.49   |
> |**w SSN (Ours)**   | 271K   | 22    | 27.69  |  26.30   |

---

> > ### Author Response · Authors · 2025-11-26
> > **Response to Reviewer TnNR (2/3)**
> >
> > > **Q2:** Can you provide more detail on the computational breakdown of SP-SSM during inference? While the paper emphasizes the overall efficiency, it would be helpful to quantify the latency contributions from superpixel generation, routing, and SSM inference separately. Also, how does the method scale on high-resolution inputs and resource-constrained devices, relative to Transformer-based SR models and MambaIR?
> >
> > **A2:** Following your suggestion, we have provided an analysis of the individual contributions of Superpixel Sampling (SPS), Mixture-of-Experts (MoE), and SSM to latency and computational efficiency, as shown in Table R2. It can be observed that SPS effectively compresses $ N$  pixels into a finite set of $ M$  superpixels, significantly reducing the computational overhead of the SSM. In addition, as shown in the table, although the introduction of MoE increases the number of parameters, it maintains high efficiency in terms of computational complexity. By using a dynamic routing mechanism to selectively select experts with different computing scales, we have effectively achieved inference acceleration while avoiding the huge computational overhead caused by full expert integration. Thus, our method can effectively achieve excellent performance at a very low computational cost.
> >
> > From Table 2 in the manuscript, it can be seen that compared to Transformer based SR models and MambaIR, our method SP-MoMamba-B has extremely low parameter count and GMACs while maintaining SOTA performance. Therefore, **our method has better scalability for deployment on high-resolution inputs and resource-constrained devices compared to MambaIR and Transofmer based SR methods.** We have revised the manuscript accordingly and included these details in Section 5.3 of the manuscript.
> >
> > **Table R2. Ablation on key components of SP-MoMamba.**
> >
> > | Method         | SSM | SPS | MoE | LMA | GatedFFN | Params | GMACs | BSD100 | Urban100 |
> > | :------------- | :-: | :-: | :-: | :-: | :------: | :----: | :---: | :----: | :------: |
> > | Baseline       | -   | -   | -   | -   | -        | 189K   | 8.4   | 27.42  | 25.55    |
> > | SP-MoMamba-T   | ✓   | -   | -   | -   | -        | 230K   | 18.7  | 27.48  | 25.64    |
> > | SP-MoMamba-T   | ✓   | ✓   | -   | -   | -        | 197K   | 16.7  | 27.54  | 25.86    |
> > | SP-MoMamba-T   | ✓   | ✓   | ✓   | -   | -        | 242K   | 16.9  | 27.59  | 26.07    |
> > | SP-MoMamba-T   | -   | -   | -   | ✓   | ✓        | 211K   | 16.6  | 27.61  | 26.14    |
> > | SP-MoMamba-T   | ✓   | ✓   | ✓   | -   | ✓        | 261K   | 19.8  | 27.63  | 26.14    |
> > | SP-MoMamba-T   | ✓   | ✓   | ✓   | ✓   | ✓        | 272K   | 22    | 27.69  | 26.40    |
> >
> > > **Q3:** In your discussion on multi-scale expert routing, could you share empirical evidence that different experts specialize in distinct spatial scales or semantic structures?
> >
> > **A3:** Following your suggestion, the decision-making process of the router at different depths is now illustrated in Table R3, with further visualization of the routing trajectory and feature maps provided in Figure 7 of the manuscript. Notably, the network showcases a diverse range of expert choices ($ \mathcal{E}_1, \mathcal{E}_2, \mathcal{E}_3$ ) across varying depths. This phenomenon can be attributed to the hierarchical feature learning nature of the architecture, aligning with our expectations. In fact, different residual groups dynamically select experts to balance global structure capture and local detail refinement. This Multi-Scale MoE design aspect provides our method with the flexibility to adaptively adjust the processing scale, a capability that ensures efficiency without compromising reconstruction quality.
> >
> > We have revised the manuscript accordingly and incorporated these details into Section 5.4 of the manuscript.
> >
> > **Table R3** **Statistics on the decisions made by SPMoMamba-T's routing function at network depth**
> >
> > |  | Urban100 |  |  |  |  | Manga109 |  |  |  |
> > |---|:---:|:---:|:---:|:---:|---|:---:|:---:|:---:|:---:|
> > |  | Residual Group | $ \mathcal{E}_1$  | $ \mathcal{E}_2$  | $ \mathcal{E}_3$ |  | Residual Group | $ \mathcal{E}_1$  | $ \mathcal{E}_2$  | $ \mathcal{E}_3$  |
> > | x2 | 1 | 90 | 8 | 2 | x2 | 1 | 8 | 91 | 10 |
> > |  | 2 | 92 | 6 | 2 |  | 2 | 92 | 16 | 1 |
> > |  | 3 | 88 | 11 | 1 |  | 3 | 10 | 8 | 91 |
> > |  | 4 | 16 | 73 | 11 |  | 4 | 9 | 89 | 11 |
> > |  | 5 | 88 | 10 | 2 |  | 5 | 82 | 23 | 4 |
> > |  | 6 | 12 | 77 | 11 |  | 6 | 92 | 16 | 1 |
> > |  | Residual Group | $ \mathcal{E}_1$  | $ \mathcal{E}_2$  | $ \mathcal{E}_3$ |  | Residual Group | $ \mathcal{E}_1$  | $ \mathcal{E}_2$  | $ \mathcal{E}_3$  |
> > | x4 | 1 | 7 | 87 | 6 | x4 | 1 | 97 | 11 | 1 |
> > |  | 2 | 89 | 8 | 3 |  | 2 | 10 | 93 | 6 |
> > |  | 3 | 11 | 85 | 4 |  | 3 | 9 | 94 | 6 |
> > |  | 4 | 10 | 89 | 1 |  | 4 | 96 | 11 | 2 |
> > |  | 5 | 3 | 18 | 79 |  | 5 | 7 | 95 | 7 |
> > |  | 6 | 85 | 14 | 1 |  | 6 | 91 | 17 | 1 |

---

> > > ### Author Response · Authors · 2025-11-26
> > > **Response to Reviewer TnNR (3/3)**
> > >
> > > > **Q4:** To better substantiate the claimed advantages, could you add comparisons or re-train baselines against recent lightweight and frequency-aware SR models—e.g., FreqFormer [1], FECAN/FECA [2], and CAMixerSR [3]—under the same training protocol and evaluation settings (×2/×4, identical data and metrics)? Reporting PSNR/SSIM as well as perceptual metrics (LPIPS/DISTS) and latency would help position your method against the current state of the art.
> > >
> > > **A4:** We sincerely appreciate your suggestion. Following your recommendation, we retrained FreFormer, FECA, and CAMixerSR according our SP-MoMamba setting. As shown in Table R4, SP-MoMamba achieves higher PSNR with reduced computational complexity compared to the state-of-the-art method, FreFormer. Furthermore, we have provided a more reasonable comparison of perceptual metrics (NIQE and BRISQUE) on real-world datasets, which further validates the superiority of our approach on perceptual metrics, as shown in Table R2 of response nJJj. We have revised the manuscript accordingly, incorporating these details into Table 4 of the manuscript, respectively.
> > >
> > > **Table R4. Comparison to lightweight SR models.**
> > >
> > > | Scale | Model               | Params (M)↓ | GMACs↓ | Set5 PSNR↑ | Set5 SSIM↑ | Set14 PSNR↑ | Set14 SSIM↑ | BSD100 PSNR↑ | BSD100 SSIM↑ | Urban100 PSNR↑ | Urban100 SSIM↑ | Manga109 PSNR↑ | Manga109 SSIM↑ |
> > > | :---- | :------------------ | :---------: | :----: | :--------: | :--------: | :---------: | :---------: | :----------: | :----------: | :------------: | :------------: | :------------: | :------------: |
> > > | ×2    | IMDN                | 604K        | 159    | 38.00      | 0.9605     | 33.63       | 0.9177      | 32.19        | 0.8996       | 32.17          | 0.9283         | 38.88          | 0.9774         |
> > > | ×2    | CAMixerSR           | 746K        | 205    | 38.23      | 0.9613     | 34.00       | 0.9214      | 32.34        | 0.9016       | 32.95          | 0.9340         | 39.32          | 0.9781         |
> > > | ×2    | FECAN-light*        | 732K        | 162    | 38.22      | 0.9614     | 34.01       | 0.9216      | 32.35        | 0.9017       | 32.89          | 0.9787         | 39.47          | 0.9784         |
> > > | ×2    | Freqformer*         | 870K        | 229    | 38.26      | 0.9615     | 34.02       | 0.9217      | 32.34        | 0.9018       | 32.94          | 0.9353         | 39.47          | 0.9789         |
> > > | ×2    | **SP-MoMamba-B (ours)** | **543K**    | **170** | **38.27**  | **0.9616** | **34.04**   | **0.9219**  | **32.38**    | **0.9022**   | **32.99**      | **0.9357**     | **40.18**      | **0.9827**     |
> > > | ×4    | IMDN                | 715K        | 41     | 32.21      | 0.8948     | 28.58       | 0.7811      | 27.56        | 0.7353       | 26.04          | 0.7838         | 30.46          | 0.9075         |
> > > | ×4    | CAMixerSR           | 765K        | 54     | 32.51      | 0.8988     | 28.82       | 0.7870      | 27.72        | 0.7416       | 26.63          | 0.8012         | 31.18          | 0.9166         |
> > > | ×4    | FECAN-light*        | 749K        | 42     | 32.53      | 0.8990     | 28.89       | 0.7879      | 27.75        | 0.7421       | 26.78          | 0.8049         | 31.47          | 0.9181         |
> > > | ×4    | FreqFormer*         | 889K        | 55     | 32.54      | 0.8991     | 28.89       | 0.7879      | 27.76        | 0.7425       | 26.73          | 0.8023         | 31.36          | 0.9178         |
> > > | ×4    | **SP-MoMamba-B (ours)** | **559K**    | **46**  | **32.56**  | **0.8992** | **28.93**   | **0.7885**  | **27.78**    | **0.7426**   | **26.76**      | **0.8030**     | **31.51**      | **0.9210**     |

---

### Official Review · Reviewer_nJJj · 2025-10-31

**Soundness:** 3
**Presentation:** 3
**Contribution:** 3
**Rating:** 6
**Confidence:** 3

**Summary:**

This paper identifies a fundamental issue in applying Mamba-based State Space Models (SSMs) to single-image super-resolution (SR): the standard 1D scanning process disrupts the 2D semantic structure of images, impairing the model's ability to capture local details. To address this, the authors propose ​SP-MoMamba, a novel framework that introduces ​superpixels​ as primary semantic units to preserve spatial relationships. The key innovation is the ​Superpixel-driven State Space Model (SP-SSM)​, which operates on superpixel regions to maintain semantic consistency. The method employs a hierarchical expert architecture to balance global semantic modeling with local detail reconstruction, aiming for an improved efficiency-performance trade-off.

**Strengths:**

* Novel and Well-Motivated Problem Formulation:​​ The paper pinpoints a fundamental yet overlooked issue in adapting Mamba for vision tasks: the "semantic disruption" caused by flattening 2D images into 1D sequences. This provides a compelling and well-justified motivation for the work.

* ​Creative Solution:​​ The core idea of leveraging superpixels as foundational units for a State Space Model (SSM) is novel. The proposed Superpixel-driven SSM (SP-SSM) addresses the identified problem by inherently preserving 2D semantic relationships.

* Comprehensive and Convincing Experiments:​​ The proposed method is rigorously compared against a wide range of existing state-of-the-art approaches across standard benchmarks. Extensive ablation studies convincingly validate the contribution of each core component, such as the SP-SSM and the hierarchical expert architecture, demonstrating their necessity and effectiveness.

**Weaknesses:**

* Insufficient Justification for Claims:​​ The assertion that previous methods (e.g., multi-directional scanning) "fail to address the fundamental problem" is somewhat strong without providing quantitative evidence or a specific metric that directly measures "semantic disruption" to support this claim convincingly.

* Limited Scope of Evaluation:​​ A primary concern is that the experimental evaluation is conducted primarily on synthetic datasets (e.g., with bicubic degradation). The paper does not demonstrate the method's performance on real-world images with complex, unknown degradations ("blind" image restoration). This omission significantly limits the claim of the method's practical applicability and generalizability, which is crucial for real-world scenarios.

**Questions:**

The paper identifies "semantic disruption" as a key limitation of Mamba-based SR. Beyond qualitative illustrations (e.g., Figure 1), are there any ​quantitative metrics​ proposed to directly measure and compare the "semantic preservation" capability of different methods?

---

> ### Author Response · Authors · 2025-11-26
> **Response to Reviewer nJJj**
>
> Thank you for your valuable feedback to help us improve our paper. We have revised our paper based on your feedback. We detail our response below and please kindly let us know if our response addresses your concerns.
> > **Q1:** Insufficient Justification for Claims: The assertion that previous methods (e.g., multi-directional scanning) "fail to address the fundamental problem" is somewhat strong without providing quantitative evidence or a specific metric that directly measures "semantic disruption" to support this claim convincingly.
>
> **A1:** To validate our claim regarding 'semantic disruption,' we analyzed the Local Attribute Map (LAM) and Diffusion Index (DI) [1] as shown in Figures 3 and 11 of the manuscript. These updates are now included in Section 3 and Appendix E.
>
> 1. **LAM Attribute (Feature Selection)**: MambaIR and MambaIRv2 exhibit sparse, localized activations, focusing largely on the single target area. In contrast, SP-MoMamba demonstrates global perception, with activated points distributed across multiple geese. This confirms our method establishes long-range dependencies, utilizing texture details from spatially separated but semantically similar objects for reconstruction.
>
> 2. **Area of Contribution**: MambaIR displays anisotropic patterns biased along scanning axes, indicating that SS2D decouples adjacent pixels. While MambaIRv2 shows slight improvement, SP-MoMamba produces an isotropic, semantic-aware heatmap. The activated region is significantly broader and aligns naturally with object boundaries, ensuring comprehensive coverage of relevant textures.
>
> 3. **Diffusion Index (DI)**: Quantitatively, SP-MoMamba achieves the **highest DI scores** (e.g., **24.36** vs. 19.28 (MambaIR)/23.91 (MambaIRv2)). This confirms that our method engages a larger range of relevant pixels, effectively preserving 2D semantic integrity compared to the limited scope of traditional scanning methods.
>
> > **Q2:** Limited Scope of Evaluation: A primary concern is that the experimental evaluation is conducted primarily on synthetic datasets (e.g., with bicubic degradation).
>
> **A2:** We appreciate your valuable suggestion. The performance evaluation on the five standard synthetic datasets was primarily intended to demonstrate the superiority of our method in terms of efficiency and reconstruction fidelity. Following your recommendation, we have conducted additional evaluations on real-world datasets, specifically RealSR and DIV2K. As shown in Table R2, the experimental results further validate the practicality and generalization capabilities of our SP-MoMamba. We have revised the manuscript accordingly and included these details in Appendix C.2.
>
> **Table R2. Real SR performance.**
>
> | Method       | NIQE $\\downarrow$ | BRISQUE $\\downarrow$| DIV2K-I (PSNR $\\uparrow$) | DIV2K-II (PSNR $\\uparrow$)|
> | :----------- | :--: | :-----: | :-----: | :------: |
> | Bicubic      | 7.65 |  58.29  |  26.30  |  25.71   |
> | SAFMN        | 7.19 |  51.39  |  26.80  |  26.77   |
> | SPIN         | 6.84 |  58.27  |  26.93  |  26.86   |
> | SeemoRe-T    | 6.53 |  45.53  |  27.07  |  27.01   |
> | SP-MoMamba-T | **6.42** |  **44.69**  |  **27.13**  |  **27.09**  |
>
> > **Q3:** The paper identifies "semantic disruption" as a key limitation of Mamba-based SR. Beyond qualitative illustrations (e.g., Figure 1), are there any quantitative metrics proposed to directly measure and compare the "semantic preservation" capability of different methods?
>
> **A3:** In addition to the qualitative explanation provided by the Local Attribute Map in Figures 3 and 11 in manuscript, we also calculated the **Diffusion Index (DI)** [1] corresponding to LAM to compare the semantic preservation capabilities of different methods. The larger the Diffusion Index, the more identical semantic pixels are involved in the restoration of the corresponding region. Therefore, from the figure, it can be seen that our method has also achieved better semantic preservation ability quantitatively than other Mamba-based methods. We have revised the paper and put these details in Section 3 of manuscript and Appendix E.
>
> [1] Gu J, Dong C. Interpreting super-resolution networks with local attribution maps[C]//Proceedings of the IEEE/CVF conference on computer vision and pattern recognition. 2021: 9199-9208.

---

### Author Response · Authors · 2025-12-03
**Comprehensive Response Summary to Reviewers' Comments**

Dear Area Chair and all reviewers:
﻿

We are grateful for the reviewers' recognition of our paper's novelty, significant contributions, and broad applicability, as well as their constructive feedback that helped us strengthen our work. Below we summarize both the positive feedback received and how we comprehensively addressed all concerns raised.
﻿

**Reviewers nJjj and TnNR** praised our identification of "semantic disruption" in Mamba-based vision models and our innovative use of superpixels as foundational units for State Space Models. TnNR specifically commended our multi-scale superpixel mixture-of-experts architecture for effectively balancing global structure modeling with fine detail restoration.
﻿

*In response to their technical concerns:*
- **Quantifying Semantic Preservation**: We introduced Local Attribute Map (LAM) and Diffusion Index (DI) metrics to quantitatively demonstrate that traditional Mamba methods suffer from semantic interruption during 1D scanning. Our approach achieves superior DI scores (24.36 vs. 23.91 (MambaIRv2)/19.28 (MambaIR)), confirming it preserves 2D semantic integrity by engaging a broader range of semantically relevant pixels than conventional scanning methods.
- **Expanded Evaluation Scope**: Beyond synthetic datasets, we conducted extensive evaluations on real-world image datasets (RealSR and DIV2K), where our method consistently outperforms existing approaches on no-reference perceptual metrics (Table 8 in the manuscript), confirming practical applicability.
- **Superpixel Clustering Implementation**: We detailed our fully differentiable end-to-end clustering approach using SSN-based soft K-means with Gumbel-Softmax, and demonstrated superior convergence stability compared to SLIC hard clustering and token pooling methods.
- **Expert Specialization Evidence**: Through depth-wise routing statistics and feature visualizations (Figure 7 of the manuscript), we confirmed that our Multi-Scale MoE dynamically selects different experts across network depths to balance global structure modeling and local detail refinement, adaptively adjusting processing scale while maintaining efficiency and reconstruction quality.
﻿
**Reviewers iSvu and L23R** praised our technical quality, "high quality English expression," and "considerable inspiration and novelty." L23R particularly valued our paper's clarity and reproducibility, noting our method achieves "SOTA performance across multiple datasets with lower computational costs" and commending our inclusion of pseudo-code and source code.
﻿

*In response to their conceptual concerns:*
- **Clarification of Innovation**: We emphasized that our work is not a simple attention-to-SSM replacement but a necessary architectural reconstruction addressing Mamba's structural limitations. Superpixels serve as fundamental scanning units that resolve the semantic interruption inherent in the traditional SSM scanning strategy. we also designed a Multi-Scale Superpixel Mixture of Experts (MSS-MoE) with dynamic routing and a Local Spatial Modulation Experts (LSME) to specifically compensate for information loss when combining superpixels with SSMs.
- **Design Necessity Verification**: Through ablation studies (Tables 3, 4, 11-15 of the manuscript), we demonstrated the critical importance of our gating mechanism and expert routing design. Alternative implementations consistently resulted in performance degradation (e.g., without gate mechanism: 32.11 vs. 32.22 PSNR on Urban100).
- **Scientific Insight Enhancement**: We articulated how superpixels and SSMs form a mutually beneficial relationship: superpixels preserve semantic continuity during sequential processing, while SSMs provide linear computational complexity that scales efficiently with superpixel representations.
- **Generalization & Limitations**: We validated our architecture's versatility on low-light enhancement tasks (Table 9 of the manuscript), achieving state-of-the-art results (24.23 (Ours) vs 23.43 (CIDNet, 2025 CVPR) and 23.31 (CWNet, 2025 ICCV) PSNR on LOLv2-Real ). We also honestly discussed limitations with extremely blurry or noisy inputs, providing concrete failure examples in the appendix.
﻿
To further strengthen our claims, we:
- Provided detailed computational breakdowns showing how superpixel sampling reduces sequence length from N to M, transforming SSM computation complexity advantageously.
- Retrained and compared against recent lightweight and frequency-domain models (Table 2 of the manuscript) under identical protocols, establishing a SOTA balance between performance and efficiency.
- Demonstrated our method's scalability advantages, with SP-MoMamba-B achieving superior PSNR with fewer parameters and lower GMACs compared to recent methods.
﻿
Our revised manuscript fully addresses all reviewers' concerns, significantly enhancing the paper's rigor and technical depth. Reviewer L23R confirmed their concerns were adequately resolved. Thank you for your consideration.

---

### Meta-Review · Area_Chair_JKaW · 2025-12-25

**Summary:**

This paper receives four marginally above the acceptance threshold and L23R satisfies the rebuttal. However, several issues raised by the reviewers are not well addressed by the authors from my point of view, including: (1) unclear motivation (nJJj), (2) evaluation on the synthetic datasets (nJJj TnNR), (3) more detailed analysis of the necessity of the modules and network design (TnNR L23R). (4) additional comparison of the latest methods (TnNR) and (5) limited novelty (iSvu L23R). As a result, this paper cannot be accepted in the current form.

**Reviewer Concerns:**

(1) unclear motivation. There is no "significantly broader activation coverage" by the proposed method than MambaIR and MambaIRv2. As a result, it cannot explain why the proposed method can solve 'semantic disruption' issue.
(2) evaluation on the synthetic datasets. Even though the authors add some evaluation on real data, the improvement is too marginal. And this will weaken the importance of the proposed method.
(3) more detailed analysis of the necessity of the modules and network design. As the authors need to add too many additional explanations and experiments according to the rebuttal, it is inappropriate to accept it currently.
(4) additional comparison of the latest methods. According to the provided results, the improvement is also too marginal.
(5) limited novelty.

**Reviewer Scores:**

NA

---

### Decision · Program_Chairs · 2026-01-26

Reject